# Lesion of striatal patches disrupts habitual behaviors and increases behavioral variability

**Jacob A. Nadel** [1,2], **Sean S. Pawelko** [1], **Della Copes-Finke** [1], **Maya Neidhart** [1], **Christopher D. Howard** [1] *

1 Neuroscience Department, Oberlin College, Oberlin, OH, United States of America, 2 Laboratory for Integrative Neuroscience, National Institute on Alcohol Abuse and Alcoholism, US National Institutes of Health, Rockville, Maryland, United States of America

* choward@oberlin.edu

## Abstract

Habits are automated behaviors that are insensitive to changes in behavioral outcomes. Habitual responding is thought to be mediated by the striatum, with medial striatum guiding goal-directed action and lateral striatum promoting habits. However, interspersed throughout the striatum are neurochemically differing subcompartments known as patches, which are characterized by distinct molecular profiles relative to the surrounding matrix tissue. These structures have been thoroughly characterized neurochemically and anatomically, but little is known regarding their function. Patches have been shown to be selectively activated during inflexible motor stereotypies elicited by stimulants, suggesting that patches may subserve habitual behaviors. To explore this possibility, we utilized transgenic mice (*Sepw1 NP67)* preferentially expressing Cre recombinase in striatal patch neurons to target these neurons for ablation with a virus driving Cre-dependent expression of caspase 3. Mice were then trained to press a lever for sucrose rewards on a variable interval schedule to elicit habitual responding. Mice were not impaired on the acquisition of this task, but lesioning striatal patches disrupted behavioral stability across training, and lesioned mice utilized a more goal-directed behavioral strategy during training. Similarly, when mice were forced to omit responses to receive sucrose rewards, habitual responding was impaired in lesioned mice. To rule out effects of lesion on motor behaviors, mice were then tested for impairments in motor learning on a rotarod and locomotion in an open field. We found that patch lesions partially impaired initial performance on the rotarod without modifying locomotor behaviors in open field. This work indicates that patches promote behavioral stability and habitual responding, adding to a growing literature implicating striatal patches in stimulus-response behaviors.

## Introduction

Organisms must optimize behavioral strategies in order to be successful in their environments. However, various strategies exist for this purpose; optimization can be rapid and strongly dependent on outcomes or slow and resistant to change. Behaviors have therefore been divided into two main categories: goal-oriented and habitual behaviors [1]. Goal-directed, or action-

**Data Availability Statement:** Data are all available in a GraphPad Prism file present in the submission.

**Funding:** This work was partially funded by a grant from Nu Rho Psi to JAN. The funder had no role in

study design, data collection and analysis, decision to publish, or preparation of the manuscript.

**Competing interests:** The authors have declared that no competing interests exist.

outcome behaviors, are sensitive to the relationship between action and outcome and are thus highly flexible. In contrast, habitual, or stimulus-response strategies, are insensitive to changes in action-outcome relationships and lead to the continued use of behaviors that do not necessarily result in positive outcomes. While habitual strategies are evolutionarily advantageous by improving cognitive efficiency, maladaptive habit formation underlies pathological states including Obsessive Compulsive Disorder [2–4], drug addiction [5–7], and Tourette's Syndrome [8]. These disorders are characterized by compulsive and maladaptive behaviors with common neuroanatomical alterations.

Habits have been studied in animal models by measuring perseverance of instrumental behaviors (e.g., lever pressing) following changes in reward value, or by measuring flexibility in responding during probes manipulating action-outcome contingency [9,10]. Distinct neural circuits supporting goal-directed and habitual behaviors have been identified using this approach [11,12]. Impairment of the dorsomedial striatum, prelimbic cortex, or orbitofrontal cortex tends to disrupt goal-directed behaviors and animals become less sensitive to changes in outcomes [13–16]. In contrast, the lateral striatum functions as a key 'habit center', as lesions of this region promote flexibility [17]. This idea is consistent with human imaging studies, which find habitual behaviors correspond to overreliance on the putamen, the primate homolog of the dorsolateral striatum [18,19]. A model has therefore been established suggesting that the dorsomedial striatum and frontal cortical inputs facilitate goal-directed actions, while the dorsolateral striatum promotes habitual behaviors [11], but see [20].

In addition to a medial-lateral divide, the dorsal striatum contains neurochemically distinct compartments: patches or striosomes compose approximately 15% of striatal volume and are surrounded by the remaining 85% of the striatum, known as the matrix [21,22]. Patches were discovered nearly 50 years ago [23], and have since been identified in the human, monkey, cat, and rodent [24]. Despite decades of research into the neuroanatomy and connectivity of striatal patches, their function remains poorly understood. Patches are heavily interconnected with limbic circuits, and they provide the only direct inhibition to midbrain dopamine neurons from the striatum [25–27], but see [28]. After repeated exposure, stimulant drugs of abuse drive expression of immediate early genes such as c-fos selectively in patches, and this expression is predictive of motor stereotypies [21,29,30]. Similarly, lesions of striatal patches reduce stimulant-induced motor stereotypies [31,32], suggesting patches may subserve compulsive behaviors. Recent work has found that pharmacological ablation of $\mu$-opioid containing neurons, which are enriched in patches, disrupts habitual responding for sucrose rewards in rats [33]. In aggregate, these studies indicate a role for patches in compulsive, habitual motor behaviors. To investigate patch involvement in habitual behaviors, we utilized transgenic mice (*Sepw1 NP67*) which express Cre-recombinase preferentially in striatal patch neurons [28,34]. We used a virus driving Cre-dependent expression of caspase 3 to selectively ablate patch neurons before training mice on a variable interval schedule of reinforcement, which has been previously used to establish habitual responding [35]. During training, we noted significantly increased day-to-day variability in response rates in lesioned mice relative to controls. Additionally, lesioning striatal patches disrupted behavioral stability across training and lesioned mice utilized a more goal-directed behavioral strategy during training. When mice were forced to omit responses in order to earn rewards, lesioned mice had diminished response rates relative to control mice, suggesting impaired habitual responding. Lesioned mice were also slightly impaired on acquisition of motor learning as assessed by performance on an accelerating, rotating balance rod (rotarod), though these mice show no generalized locomotor impairments in open field. Together, this work supports the notion that patches subserve habitual behaviors by promoting behavioral stability, an effect that cannot be solely attributed to deficits in motor control.

## Materials and methods

### Animals

All experiments were in accordance with protocols approved by the Oberlin College Institutional Animal Care and Use Committee. Mice were maintained on a 12 hr/12 hr light/dark cycle and unless otherwise noted, were provided *ad libitum* access to water and food. Experiments were carried out during the light cycle. Overall, 29 male and female Sepw1-Cre/Rosa26-EGFP mice between 2 and 5 months of age were used in this study. Sepw1-Cre mice were generously provided by Charles Gerfen (National Institutes of Health) and Nathanial Heintz (Rockefeller University). These mice show preferential Cre recombinase expression in striatal patches [28,34].

### Reagents

Isoflurane anesthesia was obtained from Patterson Veterinary (Greeley, CO, USA). Sterile and filtered phosphate buffered saline (PBS, 1X) was obtained from GE Life Sciences (Pittsburgh, PA, USA). Unless otherwise noted, all other reagents were obtained through VWR (Radnor, PA, USA).

### Viral injections

To selectively ablate striatal patches, *Sepw1 NP67* X *Rosa26-EGFP* mice were anesthetized with isoflurane (4% at 2 L/sec $O_2$ for induction, 0.5–1.5% at 0.5 L/sec $O_2$ afterward), placed in a stereotactic frame (David Kopf Instruments, Tajunga, CA, USA), and were bilaterally injected with *AAV5-flex-taCasp3-TEVp* (UNC viral vector core). Cre-dependent expression of caspase 3 has been previously shown to drive apoptosis in neurons while limiting necrosis in surrounding tissue [36]. Briefly, two burr holes were drilled above dorsal striatum (+0.9 AP, ±1.8 ML, and −2.5 DV), and a 33-gauge needle was slowly lowered to the DV coordinate over 2 minutes and held in place for 1 min prior to injections. A 5 $\mu$l syringe (Hamilton) was used to inject 0.5 $\mu$l of virus over 5 min and the needle was left in place for 5 min following injections. The needle was then slowly retracted over 5 min. Mice were sutured and received Carprofen (5 mg/kg, s.c.) as postoperative analgesia. All mice were given 3 weeks to recover before behavioral training began. Control (non-lesion control) mice underwent an identical surgical procedure but received 0.5 $\mu$l of sterile, filtered phosphate-buffered saline (PBS).

### Variable interval training

Mice were trained on a variable interval schedule to induce habitual responding ([35], see Fig 1J for experimental design). Throughout training, mice were food deprived and kept at 85% of initial weight by daily feeding of 1.5–2.5g of standard mouse chow. Operant conditioning was performed in standard operant chambers (Med Associates). Each chamber had two retractable levers on either side of a food magazine, where sucrose rewards were delivered (20% sucrose solution, 20 $\mu$l), and a house light on the opposite side of the chamber. Mice first underwent three days of continuous reinforcement training (CRF/FR1, one lever press yields one reward). At the start of the session, the house light was illuminated, and the left lever was inserted into the chamber. After 60 min or 50 rewards, the light was shut off, the lever was retracted, and the session ended. Animals that failed to obtain >10 rewards during FR1 were given an extra day of FR1 training and were excluded if they did not reach this criterion. Next, mice were trained on a variable-interval 30 task, in which they were rewarded on average 30 seconds (15–45 sec, possible intervals separated by 3 sec) contingent on lever pressing. To determine how patch lesions modified habit formation across training, lesion and control mice were divided into

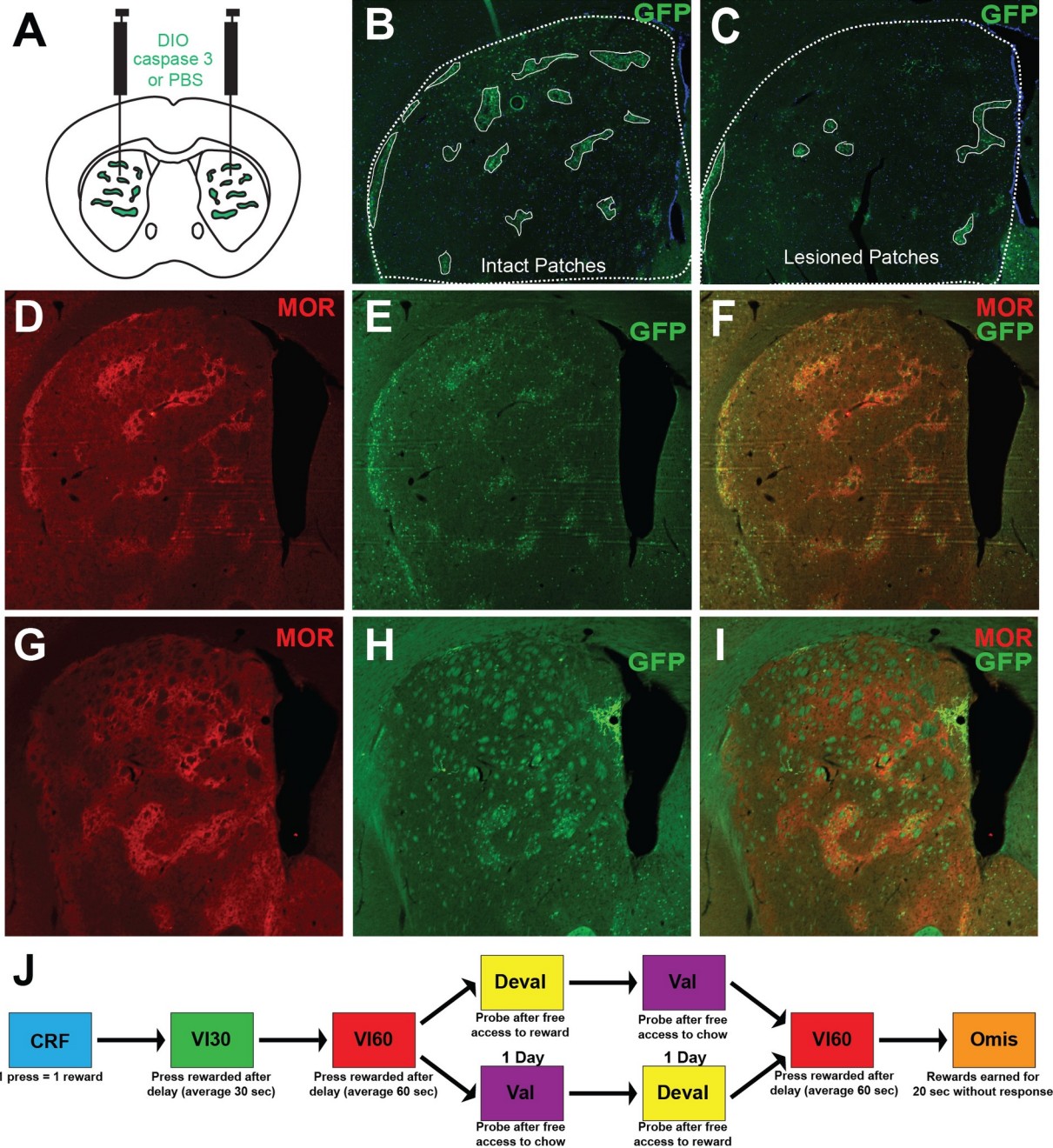

**Fig 1. Schematic of experimental design.** A. Schematic representation of injection sites in a coronal mouse brain section. Sepw1-Cre mice preferentially express Cre recombinase in striatal patches and 'exo-patches' (see text; green). AAV5-AAV-flex-taCasp3-TEVp (0.5 μl) or sterile PBS (control) was injected bilaterally into the dorsal striatum of Sepw1-Cre mice to preferentially lesion patches. B. Representative image of intact striatum of Sepw1-Cre X Rosa26-EGFP mice displaying dense GFP expression in striatal patches. Dotted line denotes border of the striatum and solid white line denotes striatal patches. C. In lesioned mice, GFP + cells are greatly reduced and striatal patches are reduced in number. D-F. Representative μ-opioid receptor (D) and GFP expression (E) and overlay (F) in intact striatum. G-I. Representative μ-opioid receptor (G) and GFP expression (H) and overlay (I) in lesioned striatum. J. Experimental design. Mice were trained to respond on a continuous reinforcement training (CRF) before beginning variable interval 30 training (VI30). This was followed by variable interval 60 (VI60) training to establish habitual responding. After training, mice experienced counterbalanced valuation/devaluation probes (Val, Deval, respectively), followed by a day of reinstatement (VI60), and two days of omission (Omis). See Methods for details of each behavioral schedule.

three groups experiencing either 3, 5, or 7 days of training on a VI60 schedule (rewarded every 60 seconds on average, ranging from 30–90 sec, possible intervals separated by 3 sec). Variable interval sessions ended after 60 min or when 50 rewards had been earned.

## Probe tests

Following completion of VI training, a devaluation test was conducted over two days. Here, mice were allowed free access to either chow (valuation) or sucrose solution (devaluation) for one hour. Immediately after, mice were given a 5-min probe test in which the lever was extended and presses were recorded, but no rewards were delivered. The order of the valued and devalued condition tests was randomized for each mouse. Mice that experienced 7 days of VI60 training only underwent a single day of devaluation after finding a significant change in response rate across probe days regardless of probe condition (see Results). One day after valuation and devaluation probe tests, mice were reinstated on the VI60 task to reestablish response rates. The following two days, mice were tested with a 60-minute omission test in which the action-outcome contingency was reversed such that mice were required to refrain from pressing the lever for 20 seconds in order to receive rewards, and pressing the lever reset the counter. Omission is a robust means of testing habitual responding [11,37], and was used to probe goal-directed control.

## Rotarod

Deficits in operant behaviors could be due to changes in habit formation or due to generalized motor deficits. Therefore, following omission tests, mice were returned to *ad libitum* access to chow for at least one week prior to assessment of motor learning. We next sought to determine how lesions of striatal patches might affect motor learning using a rotarod (Ugo Basile). Mice were initially habituated to the rod by first walking for 5 min at a slow, constant rate of 4 rpm. Lesion or control animals were then trained with four trials per day for four days where the rotarod accelerated from 3–40 rotations per min over 360 sec [38]. Each trial ended when the mouse fell from the rod or after 360 sec had elapsed. A resting period of at least 15 min separated trials. Latency to fall was recorded and compared between lesion and control groups.

## Open field

Following rotarod training, caspase-lesioned mice and controls were individually placed in a square activity chamber (42 cm wide x 42 cm long x 30 cm tall) and video-monitored from above for 30 minutes. After session completion, the distance moved, velocity, and rotation of each mouse was extracted from the video file using Ethovision (Noldus) and compared between control and lesion groups.

## Immunohistochemistry

Following the completion of behavioral experiments, mice were anesthetized with isoflurane and transcardially perfused with 0.9% saline and 4% paraformaldehyde (PFA) using a peristaltic pump or manual injection. Brains were removed and allowed to post-fix in 4% PFA at 4˚C for 24 h. Brains were then transferred to a 30% sucrose solution and returned to 4˚C. Following sinking, brains were sectioned on a freezing microtome into 25 $\mu$m sections, which were stored in a cryoprotectant solution before being washed 3X in Tris buffered saline (TBS) and blocked in 3% horse serum and 0.25% Triton X-100. Sections were then incubated in a 1:500 dilution of anti-GFP polyclonal guinea pig antibody (Synaptic Systems, cat#132–004), and/or anti-μ-opioid receptor polyclonal rabbit antibody (Immunostar, cat #24216) for 24–48 h at

4˚C on a shaker. Following incubation, sections were washed 2x15 minutes in TBS to remove excess primary antibody, then blocked for 30 minutes before incubating in Alexa Fluor® 488 AffiniPure Donkey Anti-Guinea Pig IgG (Jackson ImmunoResearch, cat#706-545-148, diluted 1:250) and/or Cy®3AffiniPure Donkey Anti-Rabbit IgG (Jackson ImmunoResearch, cat#711-165-152, diluted 1:250) for 2 hours at room temperature. Tissue was then washed 3x15 min in TBS to reduce background staining. Slices were subsequently floated in 0.1M phosphate buffer (PB) and mounted on slides. After drying, sections were covered using mounting media (Aqua-Poly/Mount, Polysciences, 18606–20) with DAPI (Sigma-Aldrich D9542; 1:1000). Tissue was visualized using a Leica DM4000B fluorescent microscope.

## Data and statistical analysis

Mean press rates and normalized press rates were compared for each probe test and reinstatement days. Devaluation probe rates for each mouse were normalized to valuation press rates (LPr; [39]) or average press rates across all VI60. Reinstatement press rates were normalized to press rates during the final day of VI60. Omission press rates were normalized to press rate during the reinstatement day following devaluation probes. Autocorrelation (lag 1) of press rates across VI60 training and cross-correlation were determined using MATLAB (R2018b, Mathworks). We intended to investigate the effects of patch lesions across different VI60 training durations (3, 5, or 7 days), but found no effect of training days across multiple task metrics, including press rates on the final day of VI60 training, and normalized response rates during valuation/devaluation probes, reinstatement day, nor omission days (p > 0.05). Therefore, we collapsed these three groups for subsequent analysis. However, due to fewer training days in the 3-day group, variability and behavioral strategy analysis was reserved for mice that received 5 or 7 days of training.

Statistical analysis was conducted using MATLAB (R2018b, Mathworks) or GraphPad Prism 7 (GraphPad). Press rates in VI30, VI60, devaluation probes, LPr, reinstatement day, and change across omission days, as well distance moved, velocity, and rotations in open field were compared between lesion and control groups with unpaired student's t-tests. Devaluation and valuation presses were compared within groups using a paired student's t-test. Efficiency was assessed by dividing number of presses or head-entries to number of rewards, which was calculated for day 1 and day 5. Day 5 efficiency was then normalized to day 1 and was compared using a one-sample t-test comparing means to 100% (no change). Similarly, press rates across omission days were compared using one-sample t-test, where day 2 press rates were normalized to day 1. Press rates across learning, probe days, omission, performance in rotarod across trials, and cross-correlations were compared using two-way repeated measures ANOVA. For ANOVAs, the Sidak's multiple comparisons test was used for post-hoc tests except for histograms and cross-correlation, where a bonferroni corrected multiple comparisons was performed. Distributions of inter-press and inter-head-entry-interval were compared using a non-parametric Two-sample Kolmogorov-Smirnov test of distribution. Finally, Pearson's Correlation was used to compare average press rate across VI60 to press rate in omission day 1. Significance was defined as $p \leq 0.05$.

## Results

### Lesion of striatal patches enhances behavioral variability

To explore patch contribution to habitual behaviors, we used Sepw1-Cre mice, which express cre-recombinase in patches [34], and an AAV encoding a modified caspase 3 virus to preferentially lesion striatal patches. Injection of AAV led to deletion of GFP+ neurons in the dorsal striatum (Fig 1A–1C). Patches have been defined by expression of μ-opioid receptor (MOR;

[21]), so we next characterized the expression of MOR in intact and lesioned tissue. GFP+ neurons preferentially aggregate in MOR-enriched striatal patches, though, as previously reported, the Sepw1 line also expresses Cre in "exo-patches," or striatal neurons outside of patches that are 'patch-like' in terms of receptor expression and development (Fig 1D–1F; [28,34]). Injection of virus encoding caspase 3 led to loss of GFP+ neurons from patches and a reduction of exo-patch neurons in both dorsomedial and dorsolateral striatum. This change was accompanied by diffuse expression of μ-opioid receptor and loss of discrete patch expression in the dorsal striatum (Fig 1G–1I). Three weeks after injection of virus ($n = 14$) or vehicle ($n = 15$), mice were trained on a variable interval schedule of reinforcement, which has been shown to induce habitual responding in mice ([35], Fig 1J). Both lesioned and control mice increased press rates across FR1, VI30, and VI60 training (two-way repeated-measures ANOVA, significant effect of day, $F_{(8,216)} = 24.9$, $p < 0.0001$) and lesioned mice were not impaired in acquisition of the task relative to controls (non-significant effect of group, $F_{(1,27)} = 0.2706$, $p = 0.6071$; non-significant interaction, $F_{(8,216)} = 1.687$, $p = 0.1028$; Fig 2A). Interestingly, across training, control mice were more consistent in their day-to-day press rates relative to patch lesioned mice. Fig 2B and 2C show the daily press rate of one mouse subtracted from the average press rate for that mouse across VI60 training in both a representative control (Fig 2B) and lesioned mouse (Fig 2C). Here, larger bars reflect increased variance across days. Indeed, across VI60 training days, lesioned mice displayed significantly increased behavioral variability in response rates (unpaired t-test, $t = 2.797$, $df = 27$, $p = 0.0094$; Fig 2D). Similarly, press rates in control mice were more predictive of press rates the following day, as they demonstrated significantly greater autocorrelation coefficients (at lag 1) relative to lesioned mice (unpaired t-test: $t = 2.144$, $df = 21$, $p = 0.0439$, Fig 2E). This suggests that lesioning patches may disrupt the stabilization of lever press rate across training, which may indicate increased behavioral flexibility. Despite this, press rates did not differ between patch lesioned or control mice in VI60 ($t = 0.3034$, $df = 27$, $p = 0.7639$, Fig 2F). Together, this suggests that lesioning striatal patches does not impair acquisition of action-outcome contingencies in VI60 training, though lesions may enhance behavioral variability across days.

## Lesion of striatal patches alters behavioral strategy and efficiency

Increased behavioral variability suggested that lesioned mice may display other differences in responding across VI training. Therefore, we plotted distributions of inter-press intervals across both groups in day 1 and day 5 of VI60 training (Fig 3A and 3B). The distribution of inter-press intervals between groups demonstrated a similar bimodal shape suggesting similar response rates between groups. Over training, control mice tend to increase their pressing around 2 sec, though the distribution does not significantly change across training (Two-sample Kolmogorov-Smirnov test, $p > 0.05$; Fig 3A), while lesioned mice tended to suppress responses at this interval (Two-sample Kolmogorov-Smirnov test, $p < 0.05$; Fig 3B). Ultimately, this resulted in a significant increase in efficiency in lesioned mice over training (one-sample t-test, $t = 2.377$, $df = 10$, $p = 0.0388$, Fig 3C), while control mice displayed no change in press:reward efficiency from day 1 to 5 (one-sample t-test, $t = 0.2779$, $df = 11$, $p = 0.7862$, Fig 3C). We next repeated this analysis for head entries into the food magazine by plotting inter-head-entry-intervals and comparing efficiency. Control mice significantly alter their distribution of inter-entry-interval, suggesting these mice increase stereotyped head entries across training at 2–4 sec intervals (Two-sample Kolmogorov-Smirnov test, $p < 0.05$; Fig 3D). On the other hand, lesioned mice tended to reduce headentries, though distributions do not significantly change across training (Two-sample Kolmogorov-Smirnov test, $p > 0.05$; Fig 3E). This resulted in a partial increase in head-entry:reward efficiency in lesioned mice (one-sample t-

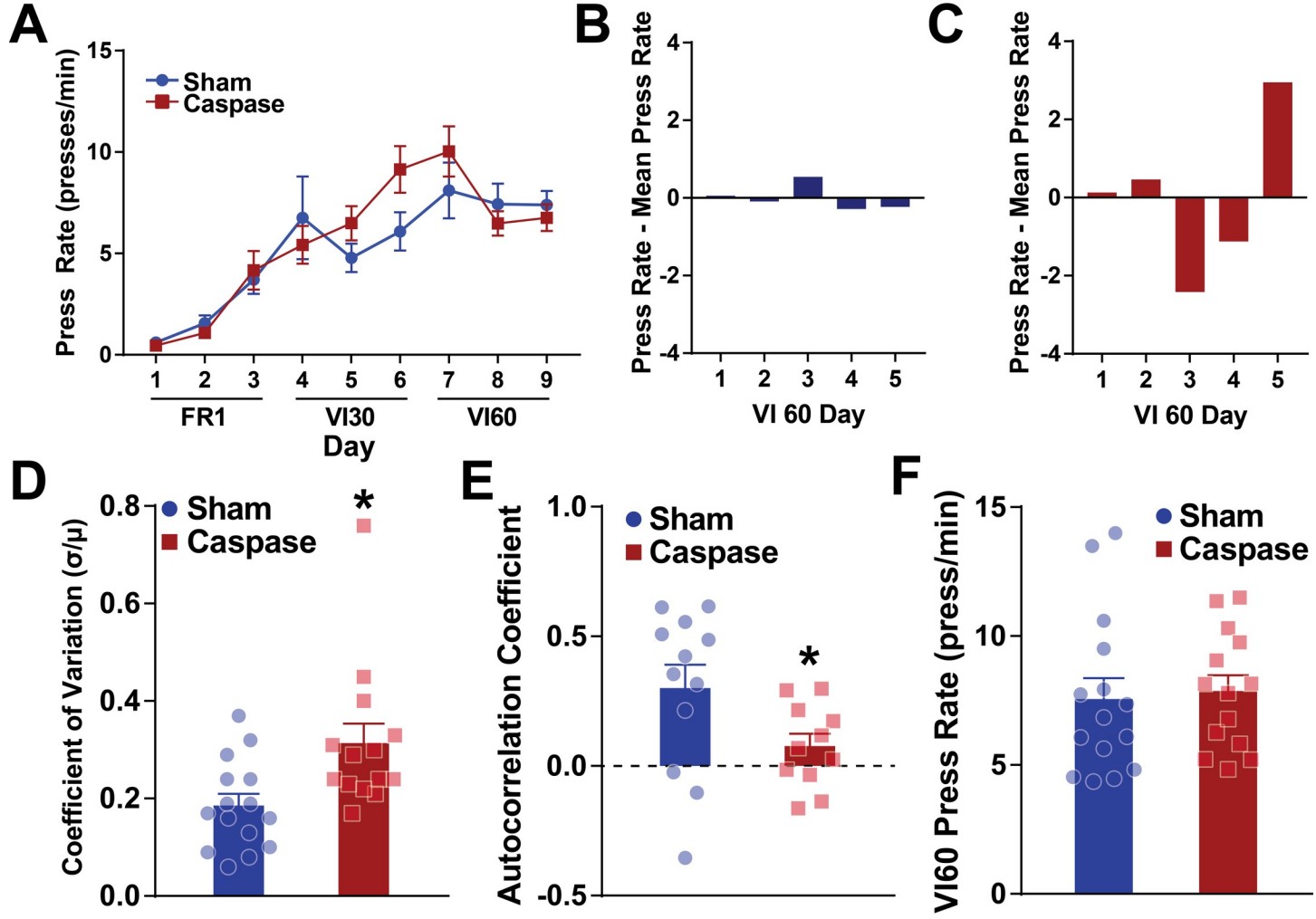

**Fig 2. Lesioning striatal patches increases response variability.** A. Across CRF (FR1), variable interval 30 (VI30), and variable interval 60 (VI60) training, lesion (red) and non-lesion control (blue) mice have similar increases in press rates. B-C. Representative day-to-day variation of press rates for a control (B) and lesioned (C) mouse. The line at 0 represents the mean press rate across all VI60 days for each respective mouse and bars represent the difference from the mean on each day. D. Coefficient of variation in press rates across VI60 training days is significantly increased in lesioned mice relative to controls. E. Autocorrelation coefficient at lag 1 is reduced in patch lesioned mice relative to controls. F. Press rates across all VI60 days are not different between lesioned and control mice. * indicates p < 0.05.

test, t = 1.917, df = 10, p = 0.0842, Fig 3F) and no change in control mice (one-sample t-test, t = 0.4354, df = 11, p = 0.6717, Fig 3F). Together, this suggests that control mice develop a less efficient strategy to obtain rewards relative to lesioned mice, potentially due to emergence of habitual, stereotyped magazine entry across learning in controls, and due to reduced pressing across learning in lesioned mice.

The differences in behavioral efficiency between lesioned and control mice may reflect differences in press/head entry patterns. That is, improved efficiency (press or entry:reward ratio) may reflect animals better learning the interval, pacing presses during the interval, and then making a head entry to determine the outcome of a press (press-check responding). On the other hand, making repeated head entries or entries followed by a press (check-press responding) may be associated with reduced efficiency by mandating multiple entries. We therefore sought to characterize the structure of responding across variable interval training for each of these groups. To characterize response patterns over time, we performed a cross-correlation

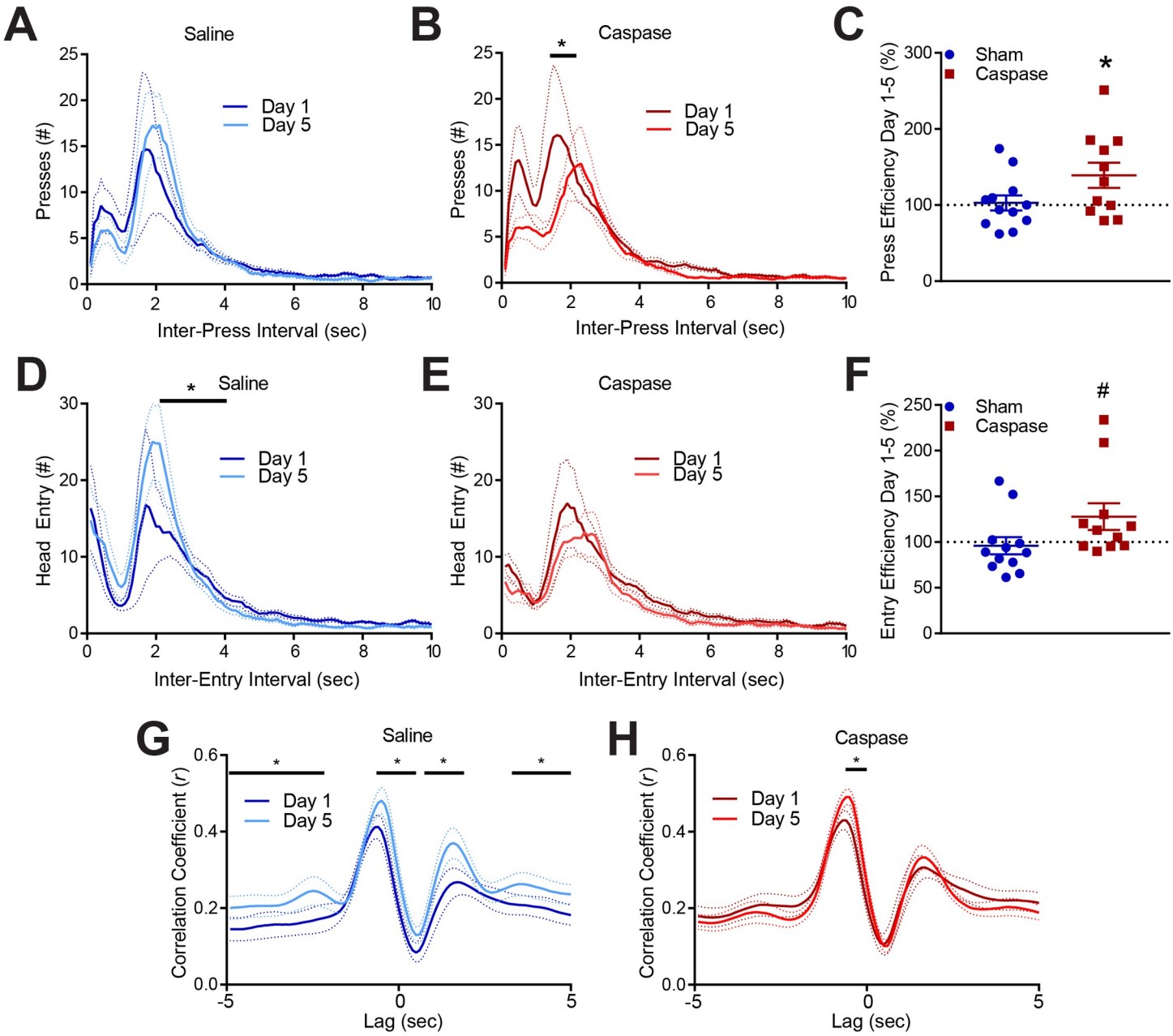

**Fig 3. Lesioned mice develop a more efficient behavioral strategy.** A-B. Distribution of inter-press interval for lesioned (A) and control mice (B) on VI60 day 1 and day 5. Solid lines represent mean and dotted lines of the same color are SEM. *indicates significantly different bins based on Kolmogorov-Smirnov test ($D_{value} > D_{critical}$) C. Lesioned mice become more efficient (change in # presses / # rewards) across training, while controls do not. D-E. Distribution of inter-entry-interval for lesioned (D) and control mice (E) on VI60 day 1 and day 5. Solid lines represent mean and dotted lines of the same color are SEM. *indicates significantly different bins based on Kolmogorov-Smirnov test ($D_{value} > D_{critical}$) F. Lesioned mice become slightly more efficient (# head-entries / # rewards) across training, while controls do not. G-H. Cross-correlation of press rate and head entry rate in 100 ms bins for control (G) and lesioned (H) mice (lags -50 to 50; see text for details). # indicates $p < 0.1$; * indicates $p < 0.05$.

analysis of presses and head-entries. Briefly, press and head-entry counts were taken across 100 ms bins for day 1 and 5 and presses were correlated to head entry at a range of intervals (lags -50 to 50). Highly correlated responding at lag 0 indicates that presses were predictive of head entries in the same 100 ms bin. Correlation at lag -50 suggests presses were predictive of

head entries 5 sec later (press-check responding), and correlation at lag 50 suggests head entries were predictive of presses 5 sec later (check-press responding). Lags between these extremes represent correlation at a shorter interval between press and entry rates. Between day 1 and 5, control mice show a change in responding with both an increase in correlation between press-check responses, and an in check-press responding (two-way repeated measures ANOVA, both factors repeated measures, significant interaction, $F_{(99,1089)} = 4.232$, $p < 0.0001$, significant bonferroni-corrected post-hoc tests shown on figure; Fig 3G). This suggests that control mice increase stereotyped press-check and check-press sequences, which is accompanied by no change in overall efficiency (Fig 3C–3F). On the other hand, lesioned mice subtly modify their responding across training, with an increased correlation in short latency press-check responding (two-way repeated measures ANOVA, both factors repeated measures, significant interaction, $F_{(99,990)} = 3.545$, $p < 0.0001$, significant bonferroni-corrected post-hoc tests shown on figure; Fig 3H). Thus, control mice increase both press-check and check-press response patterns that may indicate the emergence of reflexive, stereotyped head-entries. However, lesioned mice never increase this check-press behavior and improve their press-check responding, which is associated with increased efficiency. This improvement may suggest that patch lesioned mice maintain goal-directed responding across learning.

### Lesion of striatal patches does not disrupt devaluation press rates

Habitual behavior is operationally defined by resistance to outcome devaluation; that is, habitual organisms will continue to respond for a reinforcer even after being given free access to the reinforcer [9,10]. Thus, after the completion of training, mice were given free access to either home chow (valuation condition) or the sucrose reward they received in the operant task (devaluation condition), randomized across two days (Fig 1J). Press rates in 5 min devaluation and valuation probes did not differ in either control (paired t-test, t = 1.462, df = 11, p = 0.1717; Fig 4A) or lesioned mice (paired t-test, t = 0.6923, df = 10, p = 0.5045; Fig 4B). Further, patch lesions did not significantly impact mean devaluation press rates between groups (unpaired t-test, t = 1.362, df = 27, p = 0.1843; Fig 4C), or devaluation press rates normalized to VI60 press rates (unpaired t-test, t = 1.298, df = 27, p = 0.2054; Fig 4D). We next quantified habitual behavior by normalizing lever press rate in devaluation tests to press rates in valuation tests (LPr, see [39]) to compare the effects of reward-specific valuation to generalized satiation. Similar to devaluation tests, this metric was also not different between lesioned and control mice (unpaired t-test, t = 0.09028, df = 21, p = 0.9289; Fig 4E). However, we did observe a significant decrease in lever pressing across probe days (two-way repeated-measures ANOVA, significant effect of day, $F_{(1,21)} = 21.38$, $p < 0.0001$; Fig 4F), demonstrating that mice tended to decrease pressing across days similarly between lesion and control mice (non-significant effect of group, $F_{(1,21)} = 0.0156$, p = 0.9018, no significant interaction $F_{(1,21)} = 0.1939$, p = 0.6642). This significant decrease in press rates across subsequent probe tests is not commonly reported and indicates that *Sepw1* mice rapidly extinguish responding across subsequent probe tests. Due to the effect of day occluding any effect of probe condition, we were unable to draw conclusive inferences about the degree of habit formation from these data.

### Lesion of striatal patches alters performance during retraining and omission

Since this effect of time complicates interpretation of devaluation results, we next retrained mice with one additional day of VI6 0 (reinstatement) to reestablish high press rates. We then performed two days of omission as a further assessment of habitual responding. Here, the press contingency was reversed and mice were rewarded every 20 seconds if they refrained

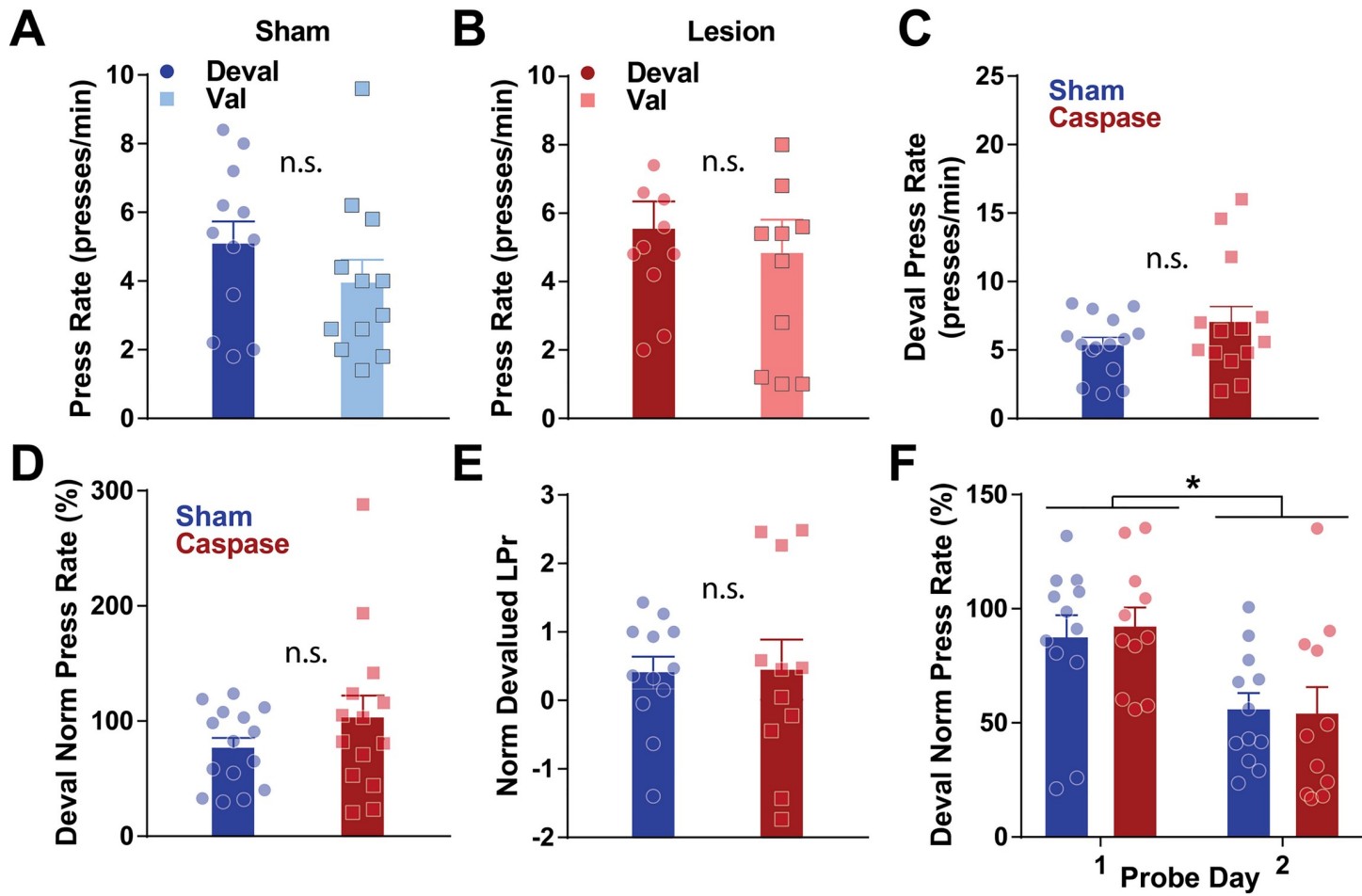

**Fig 4. Lesions of striatal patches do not change response rate during devaluation.** A-B. Press rates do not differ between devaluation and valuation probes for control (A) or lesioned mice (B). C. Press rates do not differ between lesioned (red) and control (blue) mice in devaluation D. Devaluation press rates normalized baseline press rate (average VI60 rate) is not different between lesion and control mice. E. Devaluation press rates normalized to valuation press rates (LPr, see text) did not differ between lesioned and control mice. F. Lesioned and control mice both decrease response rates across subsequent probe days.

from lever pressing, and any presses reset this timer. This approach is more efficient at extinguishing behaviors than extinction, and can be used to assess habits [11]. We first compared flexibility of mice to update their response rates to changes in task design across devaluation/valuation, reinstatement on VI60, and omission. Similar to VI60 training, lesions of striatal patches significantly increased the variance of response rates across days relative to control mice (unpaired t-test, t = 2.163, df = 27, p = 0.0396; Fig 5A), suggesting control mice maintain more consistent press rates across these probe and retraining days. Following devaluation, reinstatement to a VI60 schedule did not alter mean press rates between control and lesioned mice (unpaired t-test, t = 1.138, df = 27, p = 0.265; Fig 5B). However, lesioned mice reinstated lever pressing to a greater extent than control mice when press rates were normalized to the final day of VI60 training (unpaired t-test, t = 2.698, df = 27, p = 0.0119; Fig 5C), further indicating enhanced behavioral flexibility following patch lesions. During omission, mean press rates did not differ between lesioned and control mice (two-way repeated-measures ANOVA, non-significant effect of group or interaction, p > 0.05; Fig 5D), though both groups suppressed responding across days (two-way repeated-measures ANOVA, significant effect of time, $F_{(1,27)}$ = 31.42, p < 0.0001; Fig 5D). However, when press rates were normalized to

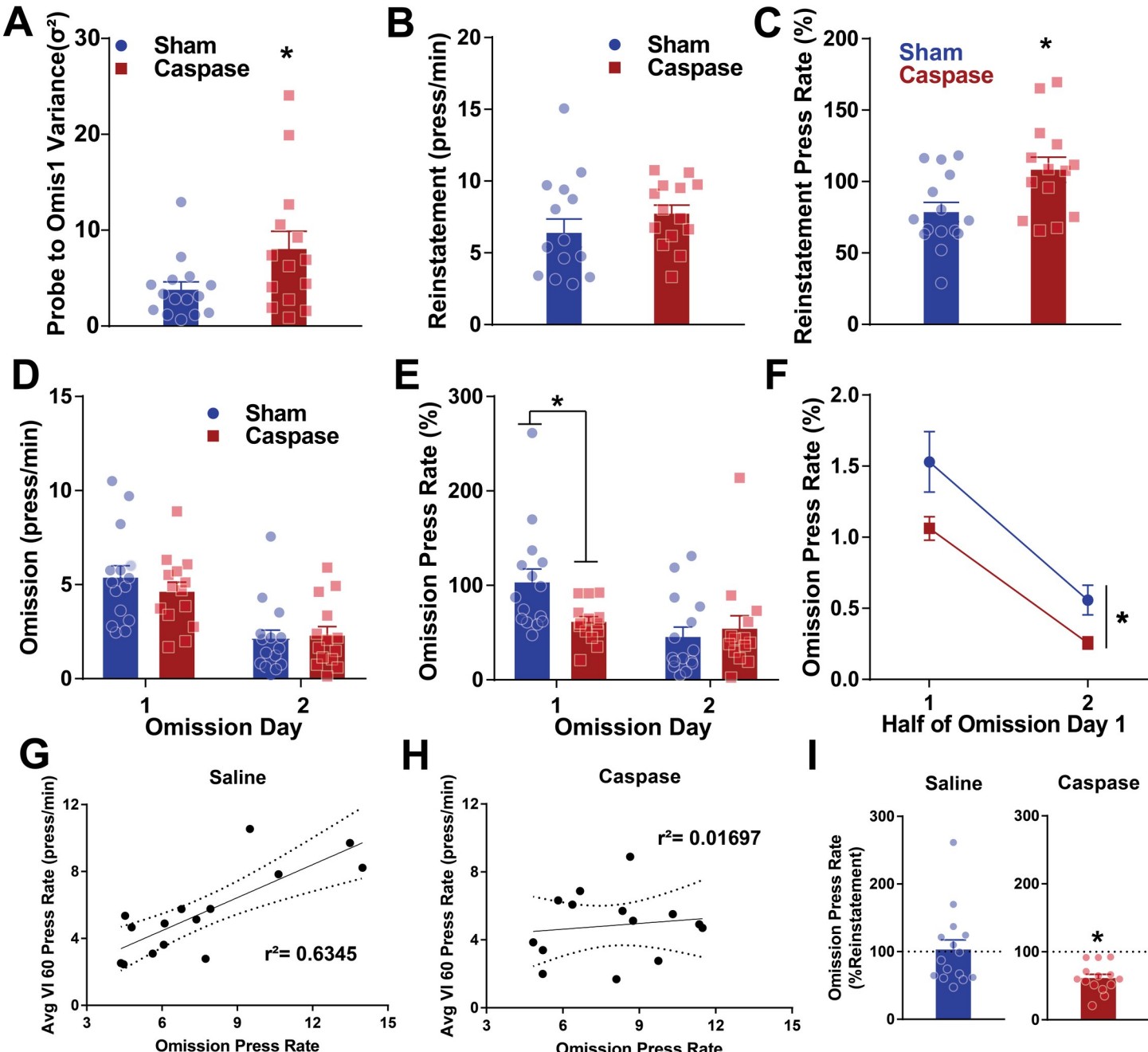

**Fig 5. Lesions of striatal patches enhance variance across probes and alter performance in omission.** A. Variance in press rate across devaluation/valuation probe, reinstatement, and omission day 1 suggest caspase lesions increase variability in press rates across days. B. During reinstatement to a VI60 schedule following devaluation/valuation probes, mean press rates do not differ between groups. C. When controlling for differences in baseline press rate, lesioned mice increase responding to a greater extent than controls during reinstatement to the VI60 schedule (data normalized to final day of VI60). D. Mice then underwent omission across two days. Mean press rates did not differ between control and lesioned mice in either day of omission. E. When controlling for differences in baseline press rate, lesioned mice rapidly reduce press rates relative to controls in day 1 of omission (data normalized to VI60 reinstatement rates). F. Press rates within the first and second half of omission day 1 suggest reduced responding in lesion mice relative to control mice. G-H. There is a significant correlation between average VI60 press rates and press rates in the first day of omission for control (G) but not lesioned mice (H). I. One sample t-test comparing change in response from reinstatement to omission suggests that control animals display habitual responding across days (left), while lesioned animals do not (right). * indicates p < 0.05.

reinstatement press rates to control for between-subject variance, lesioned mice demonstrated diminished press rates relative to control mice (two-way repeated-measures ANOVA, significant time x group interaction, $F_{(1,27)}$ = 5.17, p = 0.0311; Fig 5E), suggesting habitual responding is impaired in these mice. Post-hoc tests revealed that control mice had elevated press rates on the first day of omission compared to lesioned mice (Sidak's multiple comparisons test, Day 1, p = 0.0288). We next analyzed the press rates within the first and second halves of this first omission day. Both lesioned and control mice tended to decrease their press rate over time (two-way repeated-measures ANOVA, significant effect of time, $F_{(1,27)}$ = 83.76, p < 0.0001; Fig 5F) though lesioned mice had suppressed response rates over both halves relative to controls (significant effect of group, $F_{(1,27)}$ = 6.028, p = 0.0208, no group x time interaction, $F_{(1,27)}$ = 0.7304, p = 0.4003). Next, we assessed if average response rates across VI60 were predictive of response rates during omission to determine if mice are behaving in a stereotyped manner across time. Control mice display a significant correlation between press rate in VI60 and omission press rate (Pearson's Correlation, $r^2$ = 0.6345, p = 0.0004; Fig 5G), while this correlation was not significant for lesioned mice (Pearson's Correlation, $r^2$ = 0.01697, p = 0.6571; Fig 5H), further suggesting control mice are more consistent in press rate across days. Finally, we compared press rates across omission days to determine if mice are more habitual from omission day 1 to day 2. When press rates in omission day 2 are normalized to press rates in omission day 1, there is no significant decrease in responding in control mice between days (one-sample t-test, t = 0.2079, df = 14, p = 0.8383; Fig 5I, left) suggesting habitual responding. On the other hand, lesioned mice significantly decrease responding over time (one-sample t-test, t = 6.889, df = 13, p < 0.0001; Fig 5I, right). Together, these data suggest that control mice maintain a more stereotyped response rate across probe and retraining days, suggesting stronger habit formation in these mice.

Finally, to mirror the analysis of behavioral structure performed for VI60 training (Fig 3), we next compared the distribution of inter-press- and inter-head-entry-interval between control and lesioned mice for devaluation, valuation, and omission trials. Further, we also assessed these distributions within treatment across devaluation and valuation days. Finally, we compared the structure of behavioral responses between groups in devaluation, valuation, and omission trials. Ultimately, no significant differences were noted for these analysis (p>0.05) suggesting patch lesions did not alter behavioral strategy during probe tests.

## Lesion of striatal patches impairs motor learning, but not locomotion

Deficits in operant conditioning may be due to differences in habit formation or to generalized motor deficits. Therefore, after the completion of variable interval training, we assessed the effect of lesioning patches on motor learning using an accelerating rotarod. Mice performed four trials per day for four days, and latency to fall was measured (maximum 360 seconds; [38]). Both lesioned and control mice increased performance across days, as indicated by a significant effect of day (two-way ANOVA with multiple comparisons, main effect of day: $F_{(3,81)}$ = 49.58 p < 0.0001). However, no effect of lesion was noted across all four tested days (non-significant effect of group: $F_{(1,27)}$ = 2.119, p = 0.1570, non-significant interaction, $F_{(3,81)}$ = 1.513, p = 0.2173; Fig 6A). Within the first day of testing, lesioned and control mice improved performance (two-way repeated-measures ANOVA, significant effect of trial, $F_{(3,81)}$ = 12.54, p < 0.0001) though lesioned mice were slightly impaired relative to controls as indicated by a trending effect of group ($F_{(1,27)}$ = 3.944, p = 0.0573; non-significant interaction, p > 0.05; Fig 6B). However, by day 4, this difference was not present (two-way repeated-measures ANOVA, non-significant effect of group, $F_{(1,27)}$ = 0.1248, p = 0.7267, Fig 6C) and performance stabilized (non-significant effect of time, $F_{(3,81)}$ = 0.2656, p = 0.8500, non-significant interaction,

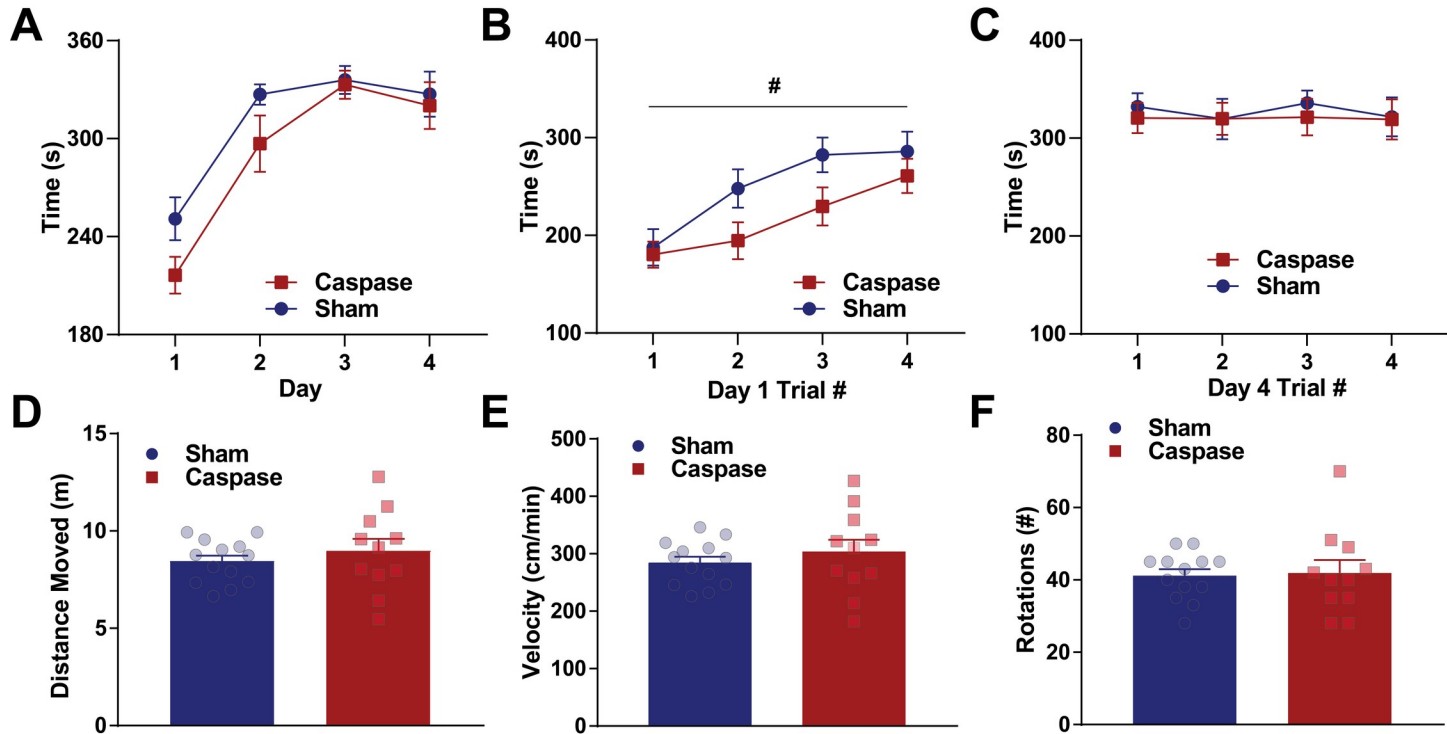

**Fig 6. Patch lesions slightly impair motor learning but not overall performance or locomotion.** A. Performance on the rotarod across days does not differ between lesioned and control mice. B. Performance during day 1 of rotarod training suggests that lesioned mice are slightly impaired relative to control mice. C. Performance on each trial within day 4 of rotarod training suggests that lesioned mice perform similarly late in training. D-F. Performance in open field suggests that lesioned mice are not different than controls in the overall distance moved (D), overall velocity (E), or in number of rotations (F). # indicates $p < 0.1$.

$p > 0.05$). This indicates that lesion of patches may partially disrupt initial motor learning, but with time, patch-lesioned mice were able to perform at the same level as control mice.

To assess overall motor activity, a subset of mice ($n = 13$ control, $n = 11$ lesion) were placed in an open field and distance moved, velocity, and rotations were quantified. We observed no differences in overall movement (unpaired t-test, $t = 0.7784$, $df = 22$, $p = 0.4446$), average velocity (unpaired t-test, $t = 0.7835$, $df = 22$, $p = 0.4417$), rotation (unpaired t-test, $t = 0.1968$, $df = 22$, $p = 0.8458$; Fig 5D–5F). These data indicate that patches may play a role in early motor learning, but that lesioning patches does not affect motor functioning.

## Discussion

Here, we investigated a role for striatal patches in habit formation and motor behaviors. To do this, we selectively lesioned patches using a Cre-dependent caspase 3 virus in *Sepw1 NP67* mice, and noted loss of striatal patches (Fig 1). Mice with patch lesions demonstrated normal learning on a variable interval task, but displayed greater day-to-day variability in response rates across training. Further, control mice developed check-press patterns of responding during training, which may reflect the development of stereotyped, habitual head entry during learning. Lesioned mice did not alter check-press behavior, but increased press-check patterns of responding, resulting in increased efficiency. Lesioned mice did not display impaired devaluation press rates, though this result is complicated by a generalized decrease in response rates across valuation and devaluation probe days. Lesioned mice also suppressed press rates faster than control mice when they were placed on an omission task, where responses had to be withheld to earn rewards, and lesioned mice were more variable in their performance across probe

and retraining days. Taken together, these results indicate that patch lesioned mice demonstrated weakened habitual behaviors and impaired behavioral stability across training and changes in task design, suggesting that striatal patches may be a key site of behavioral stability. Finally, patch lesioned mice showed slight impairment in acquisition of a new motor skill on a rotarod and no impairments in baseline locomotor activity, suggesting patches may regulate motor learning, but not motor execution *per se*, and that deficits in operant behaviors are not simply attributable to motor deficits.

In the current study, we noted that patch lesions impaired habitual responding during omission, where mice had to suppress response rates to obtain rewards (Fig 5E–5I). Omission and devaluation conditions are commonly utilized to assess habitual behaviors. However, these tasks assess different aspects of habit. Specifically, devaluation manipulates the value of the reinforcer to determine if responding persists [9,10,40]. On the other hand, omission reverses a learned action-outcome contingency, while leaving the value of the reinforcer intact. Previous work has noted that this approach results in a rapid decrease in responding, more rapid even than extinction [11,37], and it has been used to assess aspects of flexibility [39,41]. Omission is thought to reflect both the extinguishing of one behavior (e.g., lever pressing) and reinforcement of other behaviors (e.g., waiting by the food magazine), emphasizing both breakdown of an old, and learning of a new, action-outcome contingency [37]. Therefore, in the current study, while extinction was noted in both groups, lesioned mice displayed faster goal-directed control or enhanced flexibility of updating both old and new action-outcome contingencies. Indeed, while the standard view of habit formation during VI schedules is that goal-directed behaviors degrade as habits are formed, a recent study suggests that with extensive training, goal-directed behavior will eventually emerge [42]. Therefore, it is possible that, rather than disrupting habit formation, lesions of patches facilitate the emergence of goal-directed control.

Our finding of impaired habitual responses in omission is consistent with a recent study that used a conjugated cytotoxin (dermaphorin-saporin) to selectively ablate μ-opioid neurons in the striatum and that found that habit formation was impaired [33]. These findings are also consistent with studies suggesting lesions of patches impair inflexible motor stereotypies [31,32]. Jenrette et al. noted deficits in press rates when sucrose rewards were paired with lithium chloride to devalue sucrose rewards through taste aversion. However, the current study did not find a deficit in devaluation press rates when mice were provided free access to sucrose. We attribute this difference to two main factors. First, the method of devaluation (free access to reward vs. taste aversion) may not similarly devalue rewards, and it is possible that taste aversion is a more robust manipulation. Second, we noted a significant effect of probe day such that mice pressed less on day 2 regardless of probe condition (Fig 4F), indicating that the counterbalancing of days confounded any effects of probe condition. The reasons for this remain unclear, as multiple papers have successfully used this probe paradigm to assess habitual behavior [20,39]. Two factors may contribute to this finding. First, the use of home-cage chow and liquid sucrose rewards could represent an asymmetrical manipulation between devaluation and valuation probe trials, which may have impacted the results of these probe trials. However, the approach used here has been utilized in a previous study, and these mice remained sensitive to devaluation [43]. Nevertheless, the lack of difference in our probes could be attributed to potential asymmetry in consumption before devaluation and valuation probes. Another factor that might have impacted this result was length of probe trial. Our probe trial duration (5 min) greatly exceeded the delay experienced during variable interval training (30–90 sec), which might have resulted in rapid extinguishing of pressing. Other groups have used probe trials that more closely match delay times that mice experienced during training [39]. Therefore, future studies of habit using mice should be mindful of symmetry in designing valuation and devaluation probes, and in length of probes relative to variable interval delays.

It remains unclear how patches encode habitual behaviors. It is possible that disruption of striatal patches leads to over-reliance on brain circuits subserving goal-directed behaviors, including the prefrontal cortex, nucleus accumbens, and dorsomedial striatum [33]. Activity in striatal patches is tied to reward processing [44,45], and patches support intracranial self-stimulation [46], suggesting that patches have a role in reinforcement. Patch spiny projection neurons also have direct inputs to dopamine neurons [25–27] and a recent dissertation indicates they may suppress dopamine activity through $GABA_A$-mediated inward currents [47]. Lesions to patches may therefore influence spiraling basal ganglia circuits [48] by causing dysregulation of striatal dopamine release that may manifest as impaired reinforcement or disrupted decision making processes [49,50]. Indeed, dopamine signaling shifts from ventromedial to lateral striatum with extended training [51], and this process may be impacted by lesions to patches. Future studies should examine the interplay between patches and dopamine across habit formation to explore this possibility.

Alternatively, patches may mediate habitual behaviors through the endocannabinoid system in the striatum. CB1 receptors are crucial for striatal plasticity and synaptic depression [52,53], and these receptors are enriched in both striatal patches [54] and in striatal projections from the orbitofrontal cortex [14]. Indeed, the orbitofrontal cortex is thought to be key in habit and cognitive flexibility [55,56], and orbitostriatal projections are central in the transition from goal-directed to habitual strategies [13,57]. Further, knockout of CB1 receptors from orbitostriatal terminals impairs habit formation [14]. Thus, CB1 receptors are in a prime position to mediate habit-related plasticity in striatal patches. Loss of striatal patches might impair this process, which may disrupt the transfer from goal-oriented to habitual behavior.

Importantly, the use of a virus encoding caspase 3 at volumes utilized here resulted in loss of patches from the dorsal striatum spanning both medial and lateral subregions (Fig 1). Based on proposed models of striatal functioning, the medial striatum is thought to guide goal-directed behaviors [16], whereas the dorsolateral striatum and its dopamine inputs are thought to be necessary for habit formation [11,17,58], though caveats to this view have been reported [20,59]. Patches are uniformly distributed across the dorsomedial and dorsolateral striatum, forming extended compartments across anterior and posterior ends [60–62]. It is possible that medial and lateral patches have a differential role in habit formation that could reflect the larger medial-lateral divide across the striatum. Future work should investigate this possibility.

While Sepw1 NP67 mice have preferential expression of Cre recombinase in patch projection neurons, a limitation of the current work is the expression of Cre in 'exo-patch' neurons [28], resulting in the lesioning of both patch and exo-patch neurons (Fig 1). Exo-patch neurons have similar gene expression and connectivity profiles to patch neurons [28], but they fall outside of traditionally defined patches [21]. The Sepw1 NP67 line has been previously used to study patch connectivity [28,54] and activity [45]. Other recent studies have utilized alternative Cre lines to target patches, including Mash1-CreER [44], 599-CreER [63], or Oprm1-Cre [64], though each of these lines also has some off-target labeling of exo-patch or matrix neurons. Thus, while the current work suggests lesions preferentially targeting patches impair aspects of habitual behavior, we cannot rule out the contribution of exo-patch neurons in this process.

An unexpected finding from the current work was increased day-to-day behavioral variability in patch lesioned mice (Fig 2B–2E, Fig 5A). These data suggest that lesions of striatal patches may generally increase behavioral variability across days. This could suggest that patches play a general role in regulating crystallization of motor patterns, thus establishing habits. Many organisms crystalize motor patterns beyond habit formation in operant conditioning: across development, seasons, or lifespan. For example, many species of songbird show elevated variability in song production either as juveniles or during winter seasons; this variability is eventually reduced over time [65]. Indeed, the basal ganglia is thought to modulate

variability in song production in birds [66]. Moreover, spiny projection neuron distribution and patch organization differ between vocal and non-vocal songbird species [67]. Similarly, in rodents, spontaneous variation in foraging patterns are common, even following reinforcement of prior exploration (a win-shift pattern, [68,69]. Non-specific lesions of dorsal striatum impair this behavioral variability and can increase spontaneous alternation in 'win-stay' conditions, where rodents need to return to previously rewarded areas [70,71]. Future studies could investigate striatal patches as a site for stabilizing behavioral patterns in motor behaviors and reinforcement learning beyond operant conditioning.

Similarly, during habit formation, discrete behavioral elements may become chunked into larger behavioral sequences with repetition [72,73]. Indeed, as habits form, the likelihood of a given action to follow a preceding action increases [74,75]. Sensory input may therefore drive selection of concatenated behaviors once habits form, and action-outcome contingencies may be updated on a sequence-level [73,74]. It is possible that striatal patches may play a role in this aggregation of behavioral elements. Indeed, the striatum has been shown to be critical for expression of innate behavioral sequences [76,77] and learning of new behavioral sequences is particularly dependent on the lateral striatum [78]. Further, striatal neurons encode the beginning and ends of behavioral sequences as learning occurs [79,80], with differential contribution of striatal direct and indirect pathways [81]. Future studies should investigate correlates of behavioral chunking in patch neurons across habit formation.

While habitual strategies free cognitive resources and are therefore more efficient overall, goal-directed animals are sensitive to reward outcomes and might be more likely to optimize their behavioral strategy. Indeed, here, control mice begin making more stereotyped presses and head-entries and increase check-press sequences over training, establishing an inefficient, habitual checking strategy (Fig 3). On the other hand, mice with lesioned patches fail to establish this checking behavior and only improve press-check responses, resulting in an increase in efficiency. Repetitive head-entries may result in overtraining, which could enhance the establishment of inflexible responding [40]. On the other hand, the propensity of control mice to develop these behaviors may be reflective of ongoing habit formation, that is, repeated head-entries may be a marker of the establishment of habits, which is disrupted in mice with lesioned patches. Indeed, several differing views have emerged regarding why habits develop. First, it is thought that repeated pairings of behavior and reward result in habits [82]. Alternatively, tasks where the link between action and outcome is more difficult to predict drive habitual responding, explaining why random ratio schedules maintain more goal-directed responding relative to random interval schedules [40]. A related, but novel idea has been recently put forward: that tasks where animals are able to pay less attention to their responding and the outcome of behavior may drive habits [83]. Here, sham controls may be able to pay less attention to their responding due to the automacy afforded by intact patches, while lesioned mice must attend to outcomes, resulting in efficient and goal-directed behavior. Future studies utilizing variable interval schedules of reinforcement should investigate changes in responding during training that might predict habit formation.

Consistent with previous reports [84], patch lesioned mice also have slight deficits in early motor learning, but not in general movement parameters (Fig 6). Notably, minor dopamine dysfunction also leads to deficits in motor learning, but not general motor deficits [85], again raising the possibility that these deficits are partially mediated by dysfunctional dopamine regulation following patch lesions. Indeed, recent work suggests that patch lesions may drive dopamine dysfunction in the striatum, which may directly affect early motor learning [86]. Despite deficits in learning on the rotarod, it remains unlikely that motor learning is the only function of patch compartments, as our results also suggest learning of lever-pressing, locomotion, and final performance on rotarod all remain intact following patch lesion. Other studies

investigating fine motor control have found that selective inhibition of matrix neurons using DREADDs disrupts performance in reaching and grasping tasks [87]. Patch compartments have been better studied in decision making [49,50] and reward processing [44,45]. Together, this suggests that matrix neurons may regulate motor execution, whereas patch neurons regulate timing and selection of actions. Indeed, this notion is consistent with computational models [88], which hold that patches bias matrix neurons towards specific actions.

In sum, this work adds to a growing literature suggesting striatal patches support habit formation [29,33]. Lesioning patches may lead to overactivation of brain structures that support goal-oriented behaviors, including the dorsomedial striatum or prefrontal cortex. Alternatively, patch lesions may alter dopamine signaling in the striatum [25,27]. Finally, brain regions supporting inflexible behaviors have been implicated in the pathology of Obsessive Compulsive Disorder [2–4], drug addiction [5–7], and Tourette's Syndrome [8]. Future studies should investigate the contribution of striatal patches to these disease states.

## Supporting information

**S1 File. A GraphPad Prism file containing the complete data sets used in this study.**
(PZFX)

## Acknowledgments

J.A.N. was supported by the *Nu Rho Psi* Undergraduate Research Grant and the Robert Rich Student Research Grant through Oberlin College. The authors would like to thank Drs. Charles Gerfen (National Institute of Mental Health) and Nathaniel Heintz (The Rockefeller University) for generously providing Sepw1 NP67 mice. Additionally, the authors would like to thank Professor Pat Simen for fruitful discussion regarding cross correlation analysis, and Colin Dawson for discussion on analyzing variability. The authors would also like to thank Claire Geddes, Jared Smith, and Armando Salinas for discussion and feedback on the manuscript. Finally, the authors would also like to thank Lori Lindsay, Forrest Rose, Dorothy Auble, Gigi Knight, Bill Mohler, Chris Mohler and Laurie Holcomb for research support.

## Author Contributions

**Conceptualization:** Jacob A. Nadel, Christopher D. Howard.

**Data curation:** Jacob A. Nadel, Christopher D. Howard.

**Formal analysis:** Jacob A. Nadel, Sean S. Pawelko, Christopher D. Howard.

**Investigation:** Jacob A. Nadel, Sean S. Pawelko, Della Copes-Finke, Maya Neidhart.

**Methodology:** Jacob A. Nadel, Christopher D. Howard.

**Project administration:** Christopher D. Howard.

**Resources:** Christopher D. Howard.

**Software:** Jacob A. Nadel, Sean S. Pawelko, Christopher D. Howard.

**Supervision:** Christopher D. Howard.

**Validation:** Jacob A. Nadel, Christopher D. Howard.

**Visualization:** Jacob A. Nadel, Sean S. Pawelko, Christopher D. Howard.

**Writing – original draft:** Jacob A. Nadel, Christopher D. Howard.

**Writing – review & editing:** Jacob A. Nadel, Sean S. Pawelko, Della Copes-Finke, Maya Neidhart, Christopher D. Howard.

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
