## [Decision Letter · Decision Letter 0]

7 Nov 2019

PONE-D-19-28957

Lesion of striatal patches disrupts habitual behaviors and increases behavioral variability

PLOS ONE

Dear Mr. Nadel,

Thank you for submitting your manuscript to PLOS ONE. After careful consideration, we feel that it has merit but does not fully meet PLOS ONE’s publication criteria as it currently stands. Therefore, we invite you to submit a revised version of the manuscript that addresses the points raised during the review process.

In addressing the reviewers' concerns, please note my comment below regarding reviewer #2's request that you repeat the experiments.

We would appreciate receiving your revised manuscript by Dec 22 2019 11:59PM. To enhance the reproducibility of your results, we recommend that if applicable you deposit your laboratory protocols in protocols.io, where a protocol can be assigned its own identifier (DOI) such that it can be cited independently in the future. For instructions see: http://journals.plos.org/plosone/s/submission-guidelines#loc-laboratory-protocols

We look forward to receiving your revised manuscript.

Kind regards,

Jeff A Beeler

Academic Editor

PLOS ONE

Journal Requirements:

2. Our internal editors have looked over your manuscript and determined that it may be within the scope of our Neuroscience of Reward and Decision Making Call for Papers. This collection of papers is headed by a team of Guest Editors for PLOS ONE: Stephanie Groman, Satoshi Ikemoto, Jane Taylor and Robert Whelan. With this Collection we hope to bring together researchers working on a wide range of disciplines, from animal subjects research, computational approaches and patient-centered research. Additional information can be found on our announcement page: https://collections.plos.org/s/reward-and-decision-making. If you would like your manuscript to be considered for this collection, please let us know in your cover letter and we will ensure that your paper is treated as if you were responding to this call. Agreeing to be part of the call-for-papers will not affect the date your manuscript is published. If you would prefer to remove your manuscript from collection consideration, please specify this in the cover letter.

Additional Editor Comments:

Reviewer #2 asks that you repeat the experiments. As the other two reviewers did not make a similar request, the authors may respond by addressing the reviewer's concerns with either a justification or noting the issue in the manuscript. Entirely repeating the experiments is not necessary, but please address the concern.

Reviewers' comments:

Reviewer's Responses to Questions

**Comments to the Author**

1. Is the manuscript technically sound, and do the data support the conclusions?

Reviewer #1: Yes

Reviewer #2: Partly

Reviewer #3: Yes

2. Has the statistical analysis been performed appropriately and rigorously? 

Reviewer #1: Yes

Reviewer #2: No

Reviewer #3: No

3. Have the authors made all data underlying the findings in their manuscript fully available?

Reviewer #1: Yes

Reviewer #2: No

Reviewer #3: Yes

4. Is the manuscript presented in an intelligible fashion and written in standard English?

Reviewer #1: Yes

Reviewer #2: Yes

Reviewer #3: Yes

5. Review Comments to the Author

Reviewer #1: This paper reports the effects of striosome/patch lesions in the dorsolateral striatum on instrumental behavior in mice. Using a Cre driver line that allows selective lesion of patch neurons, the authors used standard procedures such as devaluation and omission to test the hypothesis that patch neurons could contribute to habitual formation. They concluded that patch lesions weakened habitual control. The experiments are well designed and novel, and on the whole the paper is well written. I do have a few questions concerning the interpretation. It'd be helpful if the authors could address these.

1. The results show that patch lesions did not affect devaluation, but increased sensitivity to imposition of the omission contingency. While both tests have been used to test for habits, they are obviously different and in principle could be dissociated. This should be explicitly discussed. The original use of the omission schedule also focuses on differential reinforcement of other behaviors, which could be relevant here (Davis and Bitterman 1971).

2. One interesting feature of the data is increased behavioral variability in the lesion mice. However, the same analysis was not performed on behavior on devaluation and omission tests. Why not? It'd be helpful to see this analysis.

3. discussion of increased variability includes the possibility of impaired reward memory consolidation. I'm not sure if I can follow the logic here. This is very briefly mentioned with no citation. Some elaboration is needed to clarify this point, or the authors should remove it.

4. The discussion of motor pattern crystalization is interesting. Dezfouli and Balleine published a series of papers examining the relationship between behavioral chunking, sequence learning, and habit formation (e.g. 2013). These should be discussed in relation to the current finding. Also, the work showing DLS contribution to sequence learning (e.g. Yin 2010) seems relevant.

Reviewer #2: This paper examines how lesions of the patch compartment in the dorsal striatum affect habitual control of instrumental performance in mice. While the main research focus has been on lateral versus medial dorsal striatum, the contributions of the patch and matrix compartments are unknown. The authors find that patch lesions induced by caspase 3 result in increased lever pressing variability and altered micro-structure of lever pressing. During selective satiety tests conducted after instrumental training, the lesion animals appear to not differ from sham animals—both seem to show instrumental insensitivity to devaluation. However, during omission tests, lesion animals show lower lever pressing rates, which suggests greater sensitivity to the action-outcome association. This is despite no apparent differences in overall motor activity, as assessed by a number of tests.

Major comments:

The method of reward devaluation is problematic. The reason for using a specific satiety procedure is to control for general satiety, but using chow as a control when the mice are food-deprived does not accomplish that. The mice are food-deprived, and they will consume more chow than sucrose solution. This makes the tests an unfair comparison, because the mice are differentially sated during the tests. If half the mice were trained to earn sucrose and the other half chow, this method would be slightly less problematic. However, given that all mice were trained to earn sucrose solution, the appropriate control is another type of solution (e.g. maltodextrin). This is an especially important point given that the authors’ goal is to elicit habits. Given that the ‘valued’ test is likely associated with increased consumption relative to the ‘devalued’ test, it’s no surprise that the authors found no difference in instrumental performance between valued and devalued test. But it’s unclear if this can be attributed to a habit or asymmetrical consumption—thus undermining the main point of the paper. I recommend running the experiment again, but using a better control for general satiety that equates consumption between valued and devalued tests.

The authors present and analyze normalized pressing rates to gauge habitual responding, but I strongly recommend presenting and analyzing the raw pressing rates for devaluation and omission tests, either instead of or in addition to the normalized rates. Raw pressing rates provide a straightforward measure of performance, while also removing any suspicion associated with normalized data.

Line 262-263: “In the context of the variable interval schedule, a single press followed by head entry is the most efficient strategy to obtain a reward, while head-entries followed by presses are less efficient”. This is confusing. To the naïve reader, it sounds like the authors are saying that on a VI, it is optimal to always check the magazine after a single lever press, and it is not optimal to check the magazine after a series of presses. Is this what the authors are suggesting? If so, it does not make much sense, and needs to be explained. It would help to present a rigorous definition of efficiency.

Line 295-296: “This finding is not consistent with prior reports”. Are the authors suggesting that the decrease in instrumental performance across two days of extinction is inconsistent with prior reports? If so, that is not true. Instrumental performance usually decreases with increased extinction experience.

For the data in Fig 5B, the results of the interaction test are not reported, and the main effect of group is not statistically significant. Yet, the authors perform a post-hoc test anyway. This seems inappropriate.

The results from the omission test are interesting, and the authors interpret the data to mean that patch lesions disrupt habit formation. An alternative interpretation that should be mentioned is that patch lesions facilitate the emergence of goal-directed control. The standard way of thinking about action control on a VI schedule is that goal-directed control appears early and then erodes with more extensive training to make way for a habit. However, a recent paper (Garr et al., 2019; DOI: 10.1037/xan0000229) has proposed a revision of this belief. Specifically, those authors argue that goal-directed control on VI schedules will eventually emerge with truly extensive training, but is also mediated by the average action-outcome interval. In light of this view, the omission data could be taken to mean that patch lesions facilitate goal-directed control, and for the sake of balance, the authors should mention it.

Minor comments:

How were the intervals distributed on the VI schedules? Also, why did the authors choose to implement a lower bound? This seems unusual to me.

There is a section titled ‘Probe trials’. I recommend changing the title to ‘Probe tests’, because the task is free-operant without any discrete trials.

Line 142-143: “Omission is a more robust means of extinguishing habitual responding…”. More robust than what?

In Fig 3A, B, D, E, there are 5 lines per plot. I assume that some of those lines represent SEM bounds, but this is not stated in the caption nor is it clearly depicted. Part of the problem is that the figures are blurry.

In Fig 4D, it’s unclear what the data are normalized to.

Line 319-320: “We next analyzed the press rates within the first and second halves of this first omission trial”. What do the authors mean by “trial”? The test is free-operant without any discrete trials, correct?

Reviewer #3: Re: PONE-D-19-28957

This present manuscript evaluates the contributions of striatal patches to instrumental learning and motor learning. Recent works have shown similar findings, using different approaches. Here, they rely on a Cre line expressed in patches, and then use a cre-dependent caspase virus to delete neurons within a patch. They find that learned lever press responding is somewhat more variable in patch-lesioned mice, and that some aspects of what has been termed habitual control are disrupted. There does not seem to be large differences in motor effects. Together, this may add novel data to the literature about how patches supporting the learning and performance of actions. There are interesting ideas about stabilizing response patterns, and in the context of Bouton’s recent work is an interesting find. A few comments made below would help to clarify and solidify the findings. In addition, there may be concerns about the stability of the Cre line used.

DATA

- I do not believe it is appropriate to use ANOVAs for analysis of the variability data. The data is bimodal in its distribution, and an assumption when using ANOVA is that the data follows a normal distribution. That being said, it is tricky implement non-parametric tests as well, but probably necessary here. As it stands, it is hard to see the argument for this being appropriate.

- Continuing with statistics, there are places where post-hoc comparisons in the absence of an a priori hypothesis are not warranted. Following up main effects within a group is fine, but comparisons of data points between groups is not appropriate if there is not a significant interaction. One such place is in the motor rod data, and the use of a post hoc to justify day 1 investigations.

- Reliance on the use of presence/absence of GFP may not be sufficient to make the claim of patch deletion. While two previous papers were cited (one of which one author was on), a quick check of those papers did not alleviate these concerns. It seems like it is quite leaky and how many cells are taken out of patches (by mu opioid staining is not clear). This doesn’t seem to be a patch selective, but just more expression within a patch.

CLARITY

-line 45, changes in action-outcomes contingencies are not determined via outcome devaluation, but by probes that examine contingency

- line 70- there is no evidence lesioned mice are “suppressing” unnecessary response.

-142, line 374, omission is robust at testing for habitual responding, and extinction is observed, but faster in goal-directed control

-line 207- press rates results are discussed, but then followed by number of lever presses, makes looking at the graphs confusing initially.

-line 288 – noticed a significant effect on press rate

- for comparisons on presence of “habitual behavior”, omission day press rate should be compared to baseline with a one-sample t-test, to show it in Sham but not caspase.

-more should be said about the distribution of patches across striatum, as this is lesions to DMS and DLS, which support seemingly different roles.

-line 453, does not seem to need a reference, or at least going to the references and seeing reviews is a bit confusing here.

6. PLOS authors have the option to publish the peer review history of their article (what does this mean?). If published, this will include your full peer review and any attached files.

Reviewer #1: No

Reviewer #2: No

Reviewer #3: No

---

## [Author Response · Author response to Decision Letter 0]

4 Dec 2019

We would like to thank the reviewers for their thoughtful and expeditious review of our manuscript. We feel that addressing these concerns will make this work stronger overall, and we are grateful for an opportunity to revise our work. As a brief overview, we have split Figure 4 into the new Figure 4 and Figure 5, we have added 13 new panels to the figures, with new statistical analyses on 10 panels, and 6 new histological panels. Further, we have included ~6 new paragraphs to the text, and many more wording changes to improve the precision of the document. Below you will find our responses to each concern with reviewer comments italicized, changes to text in bold, and our own comments in plain text. 

Reviewer #1: This paper reports the effects of striosome/patch lesions in the dorsolateral striatum on instrumental behavior in mice. Using a Cre driver line that allows selective lesion of patch neurons, the authors used standard procedures such as devaluation and omission to test the hypothesis that patch neurons could contribute to habitual formation. They concluded that patch lesions weakened habitual control. The experiments are well designed and novel, and on the whole the paper is well written. I do have a few questions concerning the interpretation. It'd be helpful if the authors could address these.

1. The results show that patch lesions did not affect devaluation, but increased sensitivity to imposition of the omission contingency. While both tests have been used to test for habits, they are obviously different and in principle could be dissociated. This should be explicitly discussed. The original use of the omission schedule also focuses on differential reinforcement of other behaviors, which could be relevant here (Davis and Bitterman 1971).

This is a very good point and we agree the text should highlight this important difference between these differential tests of habit. We have therefore included the following paragraph in the discussion:

“Omission and devaluation conditions are commonly utilized to assess habitual behaviors. However, these tasks assess different aspects of habit. Specifically, devaluation manipulates the value of the reinforcer to determine if responding persists (Adams and Dickinson 1981; Dickinson 1985). On the other hand, omission reverses a learned action-outcome contingency, while leaving the value of the reinforcer intact. Previous work has noted that this approach results in a rapid decrease in responding, more rapid even than extinction (Davis and Bitterman 1971; Yin and Knowlton 2006), and it has been used to assess aspects of flexibility (O'Hare et al. 2016; Yu et al. 2009). Omission is thought to reflect both the extinguishing of one behavior (eg. lever pressing) and reinforcement of other behaviors (e.g., waiting by the food magazine), emphasizing both breakdown of an old, and learning of a new, action-outcome contingency (Davis and Bitterman 1971). Therefore, in the current study, while extinction was noted in both groups, lesioned mice displayed faster goal-directed control or enhanced flexibility of updating both old and new action-outcome contingencies.”

2. One interesting feature of the data is increased behavioral variability in the lesion mice. However, the same analysis was not performed on behavior on devaluation and omission tests. Why not? It'd be helpful to see this analysis.

Agreed. We had not initially investigated behavioral variance across probe and omission days because the structure of the task changed. However, we have conducted this analysis and included it as the new Figure 5A. Interestingly, across probe trials, reinstatement and the first day of omission, lesioned mice are significantly more variable between days, which may reflect increased variability in these mice, or rapid updating of A-O contingencies. We then aimed to determine if press rates across VI60 training were predictive of press rates during omission. Interestingly, a significant correlation was noted for control animals. However, this correlation is not significant for lesioned mice, further suggesting rapid updating of A-O contingencies in omission, or of a general increase in variability across days. This ultimately led to a significant change in press rate in saline mice (one-sample t-test, see Reviewer 3 comment below) suggesting habitual responding on day 1 of omission, and no change in response across omission days for caspase mice, suggesting weakened habit formation. The text and figure legends have been updated accordingly. 

Next, we have analyzed distribution of inter-press-interval and inter-entry-intervals in devaluation, valuation, and the first day of omission. We have also performed cross-correlation analysis of presses and head entry to investigate structure of these behavioral sessions. Ultimately, we noted no significant differences between saline and caspase mice in valuation, devaluation, or omission. We then compared between valuation and devaluation trials within saline and caspase mice, and similarly found no significant differences. We have summarized these findings in the figure shown below (NOTE: we could not put this figure in the text box, see .pdf file of Response to Reviewers to see figure).

We opted, ultimately, not to include this as a new figure to streamline readability of the document. However, if the reviewer feels this analysis should be included as a new figure, we are happy to include it. We have included acknowledgement of this analysis in the text which can be found in results:

“Finally, to mirror the analysis of behavioral structure performed for VI60 training (Figure 3), we next compared the distribution of inter-press- and inter-head-entry-interval between control and lesioned mice for devaluation, valuation, and omission trials. Further, we also assessed these distributions within treatment across devaluation and valuation days. Finally, we compared the structure of behavioral responses between groups in devaluation, valuation, and omission trials. Ultimately, no significant differences were noted for these analysis (p>0.05) suggesting patch lesions did not alter behavioral strategy during probe tests.” 

3. discussion of increased variability includes the possibility of impaired reward memory consolidation. I'm not sure if I can follow the logic here. This is very briefly mentioned with no citation. Some elaboration is needed to clarify this point, or the authors should remove it.

We attempted to raise the possibility that some information regarding reward value could be stored in striosomes, and lesioning of patches may disrupt consolidation of memory that occurs overnight between behavioral sessions (Stickgold and Walker 2007), resulting in day-to-day variability. Indeed, striatum is thought to encode aspects of value (Knutson et al. 2009). Nevertheless, this discussion is somewhat speculative, and for clarity, we have removed this line from discussion. The text now reads: 

“This could suggest that patches play a general role in regulating crystallization of motor patterns, thus establishing habits.”

4. The discussion of motor pattern crystalization is interesting. Dezfouli and Balleine published a series of papers examining the relationship between behavioral chunking, sequence learning, and habit formation (e.g. 2013). These should be discussed in relation to the current finding. Also, the work showing DLS contribution to sequence learning (e.g. Yin 2010) seems relevant.

Agreed, this will enrich the discussion. We have included the following text”

“Similarly, during habit formation, discrete behavioral elements may become chunked into larger behavioral sequences with repetition (Graybiel 2008; Lingawi et al. 2016). Indeed, as habits form, the likelihood of a given action to follow a preceding action increases (Dezfouli and Balleine 2013; Matsumoto et al. 1999). Sensory input may therefore drive selection of concatenated behaviors once habits form, and action-outcome contingencies may be updated on a sequence-level (Dezfouli et al. 2014; Lingawi et al. 2016). It is possible that striatal patches may play a role in this aggregation of behavioral elements. Indeed, the striatum has been shown to be critical for expression of innate behavioral sequences (Berridge and Whishaw 1992; Van den Bercken, J H and Cools 1982) and learning of new behavioral sequences is particularly dependent on the lateral striatum (Yin 2010). Further, striatal neurons encode the beginning and ends of behavioral sequences as learning occurs (Cui et al. 2013; Jin et al. 2014; Jin and Costa 2015), with differential contribution of striatal direct and indirect pathways (Geddes et al. 2018). Future studies should investigate correlates of behavioral chunking in patch neurons across habit formation.”

Reviewer #2: This paper examines how lesions of the patch compartment in the dorsal striatum affect habitual control of instrumental performance in mice. While the main research focus has been on lateral versus medial dorsal striatum, the contributions of the patch and matrix compartments are unknown. The authors find that patch lesions induced by caspase 3 result in increased lever pressing variability and altered micro-structure of lever pressing. During selective satiety tests conducted after instrumental training, the lesion animals appear to not differ from sham animals—both seem to show instrumental insensitivity to devaluation. However, during omission tests, lesion animals show lower lever pressing rates, which suggests greater sensitivity to the action-outcome association. This is despite no apparent differences in overall motor activity, as assessed by a number of tests.

Major comments:

The method of reward devaluation is problematic. The reason for using a specific satiety procedure is to control for general satiety, but using chow as a control when the mice are food-deprived does not accomplish that. The mice are food-deprived, and they will consume more chow than sucrose solution. This makes the tests an unfair comparison, because the mice are differentially sated during the tests. If half the mice were trained to earn sucrose and the other half chow, this method would be slightly less problematic. However, given that all mice were trained to earn sucrose solution, the appropriate control is another type of solution (e.g. maltodextrin). This is an especially important point given that the authors’ goal is to elicit habits. Given that the ‘valued’ test is likely associated with increased consumption relative to the ‘devalued’ test, it’s no surprise that the authors found no difference in instrumental performance between valued and devalued test. But it’s unclear if this can be attributed to a habit or asymmetrical consumption—thus undermining the main point of the paper. I recommend running the experiment again, but using a better control for general satiety that equates consumption between valued and devalued tests.

The reviewer raises a valid concern and good points here. It is possible that mice might have consumed more chow than sucrose, or that consumption of chow had a more potent effect on general satiety. Accordingly, many studies have used counterbalanced liquid rewards (sucrose and maltodextrin) to accomplish reward-specific satiety in assessing habitual responding (A. Nelson and Killcross 2006; A. J. Nelson and Killcross 2013). Similarly, groups have employed differing types of solid food for with the same goal (O'Hare et al. 2016). Many studies have also used a cross-modality approach (solid and liquid rewards) similar to our own (Gremel and Costa 2013; He et al. 2016; Li et al. 2018; Renteria et al. 2018; Sieburg et al. 2019) and have found significant differences between valuation and devaluation. Indeed, the current approach using home-cage chow and liquid sucrose has been successfully employed in the literature (He et al. 2016; Li et al. 2018) Moreover, anecdotally, our own pilot experiments using C57/BL6 did not reveal statistically significant across-day decreases in lever-press rates. This point led us to suggest that rapid extinguishing across probe days may be an effect of using the Sepw1 line. 

An additional concern beyond the reviewer’s was brought to our attention recently: our variable interval schedule had, on average, 60 second delays between response and reward. However, the probe trial duration exceeded this experienced time interval. It is therefore possible that use of 5 min probe trials may be problematic, as intervals beyond the learned variable interval delay may drive rapid extinguishing of behaviors. However, probe trials exceeding delays learned during VI training are also common (eg(Jenrette et al. 2019). Nevertheless, the mouse line used, length of probe trials, and potentially asymmetric valuation and devaluation may have all culminated in our devaluation probe results. We do feel that including our data is important for the field: many habit researchers have noted across-day decreases in pressing (through word of mouth), though these results are often downplayed in the literature. Discussion of these factors will benefit this work and acknowledge shortcomings of our approach.

We have therefore added a new paragraph to Discussion to address the devaluation result: 

“Two factors may contribute to this finding. First, the use of home-cage chow and liquid sucrose rewards could represent an asymmetrical manipulation between devaluation and valuation probe trials, which may have impacted the results of these probe trials. Many studies have used counterbalanced liquid rewards (sucrose and maltodextrin) to accomplish reward-specific satiety in assessing habitual responding (A. Nelson and Killcross 2006; A. J. Nelson and Killcross 2013). Similarly, groups have employed differing types of solid food for with the same goal (O'Hare et al. 2016). However, the approach used here has been utilized elsewhere (He et al. 2016; Li et al. 2018) and, more generally, the combination of liquid and solid reinforcers used across devaluation and valuation trials is common (Gremel and Costa 2013; Gremel et al. 2016; He et al. 2016; Li et al. 2018; Renteria et al. 2018). Nevertheless, the lack difference in our devaluation and valuation trials could be attributed to this asymmetry in reward, which Sepw1 mice may be particularly sensitive to. Another factor that might have impacted this result was length of probe trial. Our probe trial duration (5 min) greatly exceeded the delay experienced during variable interval training (30-90 sec), which might have resulted in rapid extinguishing of pressing. Other groups have used probe trials that more closely match delay times that mice experienced during training (O'Hare et al. 2016). Therefore, future studies of habit using mice should be mindful of symmetry in designing valuation and devaluation probes, and in length of probes relative to variable interval delays.” 

The authors present and analyze normalized pressing rates to gauge habitual responding, but I strongly recommend presenting and analyzing the raw pressing rates for devaluation and omission tests, either instead of or in addition to the normalized rates. Raw pressing rates provide a straightforward measure of performance, while also removing any suspicion associated with normalized data.

This is also a good suggestion. The use of normalized press rates is a common means of controlling for variable baseline press rates between individuals eg. (Gremel and Costa 2013; Hilario et al. 2007; O'Hare et al. 2016; Renteria et al. 2018; Yin et al. 2004). Our intent was to model this approach, particularly with a dataset with significant variability, as well as to streamline readability of the document. We have nevertheless split Figure 4 into Figure 4 and Figure 5, and provided raw press rates for Devaluation, Reinstatement, and Omission, as well as including additional variance analysis per reviewer #1. The text has been amended to support this change.

Press rates in devaluation, reinstatement, and omission do not differ between groups, suggesting the largest effects are found across days. Together, these results suggest that patch-lesioned mice are more flexible in updating their action-outcome contingencies to new reinforcement schedules across days, which is consistent with our overarching argument: that patch lesions make animals more flexible, either by impacting habit formation or by facilitating goal-directed responding.

Line 262-263: “In the context of the variable interval schedule, a single press followed by head entry is the most efficient strategy to obtain a reward, while head-entries followed by presses are less efficient”. This is confusing. To the naïve reader, it sounds like the authors are saying that on a VI, it is optimal to always check the magazine after a single lever press, and it is not optimal to check the magazine after a series of presses. Is this what the authors are suggesting? If so, it does not make much sense, and needs to be explained. It would help to present a rigorous definition of efficiency.

We apologize for the lack of clarity here. Our attempt was to suggest that the difference in efficiency between these groups may reflect some difference in press/head entry patterns. If the average variable interval is well learned, a mouse that is behaving optimally may wait, press a minimal number of times, and immediately seek feedback following pressing by making a head entry. On the other hand, if mice continually make head entries that did not follow a press, it is less likely these head entries will be rewarded (resulting in reduced efficiency). The language was, however, imprecise and we have attempted to clarify this point by reworded this section. We have also including our definition of efficiency, and attempted to make the language more clear by implementing a simplified discussion of the meaning of the cross-correlation data. We have updated the language in Discussion to match these edits:

“The differences in behavioral efficiency between lesioned and control mice may reflect differences in press/head entry patterns. That is, improved efficiency (press or entry:reward ratio) may reflect animals better learning the interval, pacing presses during the interval, and then making a head entry to determine the outcome of a press (press-check responding). On the other hand, making repeated head entries or entries followed by a press (check-press responding) may be associated with reduced efficiency by mandating multiple entries. We therefore sought to characterize the structure of responding across variable interval training for each of these groups… Correlation at lag -50 suggests presses were predictive of head entries 5 sec later (press-check responding), and correlation at lag 50 suggests head entries were predictive of presses 5 sec later (check-press responding). Lags between these extremes represent correlation at a shorter interval between press and entry rates. Between day 1 and 5, control mice show a change in responding with both an increase in correlation between press-check responses, and an in check-press responding (two-way repeated measures ANOVA, both factors repeated measures, significant interaction, F(99,1089) = 4.232, p < 0.0001, significant bonferroni-corrected post-hoc tests shown on figure; Fig 3G). This suggests that control mice increase stereotyped press-check and check-press sequences, which is accompanied by no change in overall efficiency (Fig 3 C + F). On the other hand, lesioned mice subtly modify their responding across training, with an increased correlation in short latency press-check responding (two-way repeated measures ANOVA, both factors repeated measures, significant interaction, F(99,990) = 3.545, p < 0.0001, significant bonferroni-corrected post-hoc tests shown on figure; Fig 3H). Thus, control mice increase both press-check and check-press response patterns that may indicate the emergence of reflexive, stereotyped head-entries. Lesioned mice never increase this check-press behavior and improve their press-check responding, which is associated with increased efficiency. This improvement may suggest that patch lesioned mice maintain goal-directed responding across learning.”

Line 295-296: “This finding is not consistent with prior reports”. Are the authors suggesting that the decrease in instrumental performance across two days of extinction is inconsistent with prior reports? If so, that is not true. Instrumental performance usually decreases with increased extinction experience.

We apologize for lack of clarity here. While extinction has been repeatedly shown to drive decreases in response rate (Davis and Bitterman 1971; Yin and Knowlton 2006), significant decreased response rates across two probe trials are not commonly reported. We have amended this statement accordingly:

“This significant decrease in press rate across subsequent probe trials is not commonly reported.”

For the data in Fig 5B, the results of the interaction test are not reported, and the main effect of group is not statistically significant. Yet, the authors perform a post-hoc test anyway. This seems inappropriate.

We have amended this by removing post hoc analysis to address this concern and those of Reviewer #3. Additionally, we have softened the language in the text by adding the word ‘slightly’ to describe impairments in rotarod, which better reflects the trending effect of lesion in day 1 of rotarod training. We have updated the figure, figure legend, and text accordingly. Similarly, the new figure 5F has also had post hoc tests removed as the interaction was not significant. A significant effect of lesion is now shown on the figure. 

The results from the omission test are interesting, and the authors interpret the data to mean that patch lesions disrupt habit formation. An alternative interpretation that should be mentioned is that patch lesions facilitate the emergence of goal-directed control. The standard way of thinking about action control on a VI schedule is that goal-directed control appears early and then erodes with more extensive training to make way for a habit. However, a recent paper (Garr et al., 2019; DOI: 10.1037/xan0000229) has proposed a revision of this belief. Specifically, those authors argue that goal-directed control on VI schedules will eventually emerge with truly extensive training, but is also mediated by the average action-outcome interval. In light of this view, the omission data could be taken to mean that patch lesions facilitate goal-directed control, and for the sake of balance, the authors should mention it.

We absolutely agree, lesions of patches could reflect increases in goal-directed control, which may reflect rapid updating of response rates across omission and probe tests. We have therefore added a section to Discussion to address this possibility: 

“Indeed, the standard view of habit formation during VI schedules is that goal-directed behaviors degrade as habits are formed, a recent study suggests that with extensive training, goal-directed behavior will eventually emerge (Garr et al. 2019). Therefore, it is possible that, rather than disrupting habit formation, lesions of patches facilitate the emergence of goal-directed control.”

Minor comments:

How were the intervals distributed on the VI schedules? Also, why did the authors choose to implement a lower bound? This seems unusual to me.

VI30 spanned 15-45 sec while VI60 spanned 30-90 sec. Intervals at each trial were randomly selected from a list of possible intervals, each separated by 3 sec (30, 33, 36, etc.). Lower and upper bounds were selected to reflect time intervals across a wide range of times to mirror probability-based random interval schedules. This approach led to escalating press rates similar to previous reports (Fig 2A) (Hilario et al. 2007). To clarify this point we have expanded details in Methods:

“30 seconds (15-45 sec, possible intervals separated by 3 sec)... (rewarded every 60 seconds on average, ranging from 30-90 sec, possible intervals separated by 3 sec)”

There is a section titled ‘Probe trials’. I recommend changing the title to ‘Probe tests’, because the task is free-operant without any discrete trials.

Done. Further, we have converted the term “trial” to read test when referring to probe tests, we have deleted the word trial when referencing omission days, and we have only left the word trial when referring to rotarod trials that took place on the same day.

Line 142-143: “Omission is a more robust means of extinguishing habitual responding…”. More robust than what?

This line has been amended to address this and the concern raised by reviewer #3 below:

“Omission is a robust means of testing habitual responding (Davis and Bitterman) [19], and was used to probe goal-directed control”

In Fig 3A, B, D, E, there are 5 lines per plot. I assume that some of those lines represent SEM bounds, but this is not stated in the caption nor is it clearly depicted. Part of the problem is that the figures are blurry.

This may be partly due to compression during submission. Our .PNGs were exported at 300 dpi, which should be standard, publication quality resolution, though we see that the images in the pdf are heavily compressed. There are 6 lines per figure, and as the reviewer assumed, the dotted lines are SEM. We have updated figure legends accordingly:

“Solid lines represent mean and dotted lines of the same color are SEM”

In Fig 4D, it’s unclear what the data are normalized to.

Information about normalization is located in Data Analysis (“Reinstatement press rates were normalized to press rates during the final day of VI60”), but we have amended the figure legend to be more clear:

“Lesioned mice increased responding to a greater extent than controls during reinstatement to the VI60 schedule (data normalized to final day of VI60).”

Line 319-320: “We next analyzed the press rates within the first and second halves of this first omission trial”. What do the authors mean by “trial”? The test is free-operant without any discrete trials, correct?

This was imprecise throughout the text. We have converted the term “trial” to read test when referring to probe tests, we have deleted the word trial when referencing omission days, and we have only left the word trial when referring to rotarod trials that took place on the same day. 

Reviewer #3: Re: PONE-D-19-28957

This present manuscript evaluates the contributions of striatal patches to instrumental learning and motor learning. Recent works have shown similar findings, using different approaches. Here, they rely on a Cre line expressed in patches, and then use a cre-dependent caspase virus to delete neurons within a patch. They find that learned lever press responding is somewhat more variable in patch-lesioned mice, and that some aspects of what has been termed habitual control are disrupted. There does not seem to be large differences in motor effects. Together, this may add novel data to the literature about how patches supporting the learning and performance of actions. There are interesting ideas about stabilizing response patterns, and in the context of Bouton’s recent work is an interesting find. A few comments made below would help to clarify and solidify the findings. In addition, there may be concerns about the stability of the Cre line used.

DATA

- I do not believe it is appropriate to use ANOVAs for analysis of the variability data. The data is bimodal in its distribution, and an assumption when using ANOVA is that the data follows a normal distribution. That being said, it is tricky implement non-parametric tests as well, but probably necessary here. As it stands, it is hard to see the argument for this being appropriate.

This is a good point. Further, these data are distributions of inter-response-intervals, and an analysis of distribution is a more appropriate approach than ANOVA. We therefore reanalyzed this data set utilizing a two-sample Kolmogorov-Smirnov test of distribution, which is non-parametric, and which compares distribution shape. We find that distribution of inter-press-intervals changes across training in lesioned mice, but not controls, lesioned mice increase efficiency (responses/reward) by decreasing stereotyped pressing across training. On the other hand, control mice do not increase efficiency, this is accompanied by no change in IPI distribution. Caspase mice do not significantly alter their distribution of inter-head-entry-interval across learning, though they display a trend for increased efficiency. Controls significantly alter their distribution of inter-head-entry-intervals by increasing stereotyped entries, which is associated with no increase in efficiency over training.

This analysis has replaced the previous ANOVA results in the text and figure:

“Over training, control mice tend to increase their pressing around 2 sec, though the distribution does not significantly change across training (Two-sample Kolmogorov-Smirnov test, p>0.05; Fig 3A) , while lesioned mice tended to suppress responses at this interval (Two-sample Kolmogorov-Smirnov test, p<0.05, Fig 3B).”

“Control mice significantly alter their distribution of inter-entry-interval, suggesting these mice increase stereotyped head entries across training at 2-4 sec intervals (Two-sample Kolmogorov-Smirnov test, p<0.05; Fig 3D). On the other hand, lesioned mice tended to reduce headentries, though distributions do not significantly change across training (Two-sample Kolmogorov-Smirnov test, p>0.05; Fig 3E). This resulted in a partial increase in head-entry:reward efficiency in lesioned mice (one-sample t-test, t = 1.917, df = 10, p =0.0842, Fig 3F) and no change in control mice (one-sample t-test, t = 0.4354, df = 11, p =0.6717, Fig 3F). Together, this suggests that control mice develop a less efficient strategy to obtain rewards relative to lesioned mice, potentially due to emergence of habitual magazine entry across learning in controls, and due to reduced pressing across learning in lesioned mice.”

- Continuing with statistics, there are places where post-hoc comparisons in the absence of an a priori hypothesis are not warranted. Following up main effects within a group is fine, but comparisons of data points between groups is not appropriate if there is not a significant interaction. One such place is in the motor rod data, and the use of a post hoc to justify day 1 investigations.

The reviewer is also correct on this point. We have removed this post hoc analysis to address this concern, and that of reviewer #2 above. Additionally, we have softened the language in the text by adding the word ‘slightly’ to describe impairments in rotarod, which better reflects the trending effect of lesion in day 1 of rotarod training. We have updated the figure, figure legend, and text accordingly. Additionally, this was also true of old figure 4F/ new figure 5F. The post hoc tests have been omitted and group effects are now indicated on this panel.

- Reliance on the use of presence/absence of GFP may not be sufficient to make the claim of patch deletion. While two previous papers were cited (one of which one author was on), a quick check of those papers did not alleviate these concerns. It seems like it is quite leaky and how many cells are taken out of patches (by mu opioid staining is not clear). This doesn’t seem to be a patch selective, but just more expression within a patch.

The reviewer raises a valid concern. Indeed, in our own histological analysis of this line, we have noted Cre expression outside of patches, which can be seen in Figure 1. The work the reviewer mentions characterized these so-called ‘exo-patch’ neurons (Cre+ neurons outside of traditionally defined patches) and found similarities in µ-opioid receptor expression, D1/D2 receptor expression, and embryological development between patches and exo-patches, suggesting they may be similar in nature (supplemental figures 1-3)(Smith et al. 2016). Further, the Sepw1 line has been shown to have preferential output to SNc dopamine neurons, consistent with established patch architecture (Gerfen et al. 2013; Smith et al. 2016). Nevertheless, the language in the current work can be softened to acknowledge the possibility that our lesions likely occurred in patch and “exo-patch” SPNs. Specifically, use of the term “selective” when describing Cre-expression or lesion to patches has been converted to “preferential” throughout the text. 

The reviewer’s point about GFP vs MOR expression is also duly noted. We have now performed this stain and reworked Figure 1 to include representative images of MOR expression in intact and lesioned striatum. We see loss of GFP+ cells in dorsal striatum, and diffuse MOR expression in this region that is not well defined into discrete patches (though ventral patches are still present). This stain suggests that both GFP and MOR expression are disrupted following injection of virus encoding caspase 3. The text has been amended to support this change in the figure:

“…an AAV encoding a modified caspase 3 virus to preferentially lesion striatal patches. Injection of AAV led to deletion of GFP+ neurons in the dorsal striatum (Fig 1A-C). Patches have been defined by expression of µ-opioid receptor (MOR; Crittenden and Graybiel, 2011), so we next characterized the expression of MOR in intact and lesioned tissue. GFP+ neurons preferentially aggregate in MOR-enriched striatal patches, though, as previously reported, the Sepw1 line also expresses Cre in “exo-patches”, or striatal neurons outside of patches that are ‘patch-like’ in terms of receptor expression and development (Fig 1 D-F; Smith et al. 2016). Injection of virus encoding caspase 3 led to loss of GFP+ neurons from patches and a reduction of exo-patch neurons in both dorsomedial and dorsolateral striatum. This change was accompanied by diffuse expression of µ-opioid receptor and loss of discrete patch expression in the dorsal striatum (Fig 1G-I).”

 CLARITY

-line 45, changes in action-outcomes contingencies are not determined via outcome devaluation, but by probes that examine contingency

True, we have updated the text to be more precise: 

“Habits have been studied in animal models by measuring perseverance of instrumental behaviors (e.g. lever pressing) following changes in reward value [9,10], or by measuring flexibility in responding during probes manipulating action-outcome contingency (Davis and Bitterman; Yin and Knowlton 06).”

- line 70- there is no evidence lesioned mice are “suppressing” unnecessary response.

This has been omitted: 

“Additionally, lesioning striatal patches disrupted behavioral stability across training and lesioned mice utilized a more goal-directed behavioral strategy during training.”

-142, line 374, omission is robust at testing for habitual responding, and extinction is observed, but faster in goal-directed control

142: “Omission is a robust means of testing habitual responding (Davis and Bitterman) [19], and was used to probe goal-directed control”

374: “Therefore, in the current study, while extinction was noted in both groups, lesioned mice displayed faster goal-directed control or enhanced flexibility of updating both old and new action-outcome contingencies.”

-line 207- press rates results are discussed, but then followed by number of lever presses, makes looking at the graphs confusing initially.

The reviewer makes a good point here, the use of press # is problematic as length of behavioral session changes between days. We have converted this figure to press rate and updated the text accordingly. It now reads:

“Figure 2B+C show the daily press rate of one mouse subtracted from the average press rate for that mouse across VI60 training in both a representative control (Fig 2B) and lesioned mouse (Fig 2C).”

-line 288 – noticed a significant effect on press rate

“Patch lesions did not significantly impact press rates during devaluation tests”

- for comparisons on presence of “habitual behavior”, omission day press rate should be compared to baseline with a one-sample t-test, to show it in Sham but not caspase.

This has now been performed and is included as then new Figure 5I. Relative to basline pressing (Reinstatement press rate), sham mice persist in press rates during the first day of omission, indicating habitual behavior. On the other hand, caspase lesioned mice have significantly reduced press rates in omission relative to baseline, suggesting impaired habitual responding. The text and figure legends have been updated to reflect this change. This comment also inspired us to investigate press rates across VI60 to determine if they reflected press rates in omission. Interestingly, there is a significant correlation in control, but not lesioned mice, suggesting lesioned mice rapidly update A-O contingencies in omission (Figure 5G-H).

-more should be said about the distribution of patches across striatum, as this is lesions to DMS and DLS, which support seemingly different roles.

We agree. Indeed, when we presented parts of this work at the recent Society for Neuroscience conference, this question was the most commonly asked about the data set. To address this, we have added a brief paragraph to Discussion exploring different functions of medial and lateral patches:

“Importantly, use of a virus encoding caspase 3 at volumes utilized here resulted in loss of patches from the dorsal striatum spanning both medial and lateral subregions (Figure 1). Based on proposed models of striatal functioning, the medial striatum is thought to guide goal-directed behaviors (Yin et al. 2005), whereas the dorsolateral striatum and its dopamine inputs are thought to be necessary for habit formation (Faure et al. 2005; Yin et al. 2004; Yin and Knowlton 2006), though caveats to this view have been reported (Malvaez et al. 2018; Okada et al. 2014). Patches are uniformly distributed across the dorsomedial and dorsolateral striatum, forming extended compartments across anterior and posterior ends (Desban et al. 1993; Johnston et al. 1990; Morigaki and Goto 2016). It is possible that medial and lateral patches have a differential role in habit formation that could reflect the larger medial-lateral divide across the striatum. Future work should investigate this possibility.” 

-line 453, does not seem to need a reference, or at least going to the references and seeing reviews is a bit confusing here.

This citation has been removed.

References

Adams C. D. and Dickinson A. (1981) Instrumental responding following reinforcer devaluation. The Quarterly Journal of Experimental Psychology Section B. 33, 109-121.

Berridge K. C. and Whishaw I. Q. (1992) Cortex, striatum and cerebellum: control of serial order in a grooming sequence. Exp. Brain Res. 90, 275-290.

Cui G., Jun S. B., Jin X., Pham M. D., Vogel S. S., Lovinger D. M. and Costa R. M. (2013) Concurrent activation of striatal direct and indirect pathways during action initiation. Nature. 494, 238-242.

Davis J. and Bitterman M. E. (1971) Differential reinforcement of other behavior (DRO): a yoked-control comparison. J. Exp. Anal. Behav. 15, 237-241.

Desban M., Kemel M. L., Glowinski J. and Gauchy C. (1993) Spatial organization of patch and matrix compartments in the rat striatum. Neuroscience. 57, 661-671.

Dezfouli A. and Balleine B. W. (2013) Actions, action sequences and habits: evidence that goal-directed and habitual action control are hierarchically organized. PLoS Comput. Biol. 9, e1003364.

Dezfouli A., Lingawi N. W. and Balleine B. W. (2014) Habits as action sequences: hierarchical action control and changes in outcome value. Philos. Trans. R. Soc. Lond. B. Biol. Sci. 369, 10.1098/rstb.2013.0482.

Dickinson A. (1985) Actions and Habits: The Development of Behavioural Autonomy. Philosophical Transactions of the Royal Society of London.Series B, Biological Sciences. 308, 67-78.

Faure A., Haberland U., Conde F. and El Massioui N. (2005) Lesion to the nigrostriatal dopamine system disrupts stimulus-response habit formation. J. Neurosci. 25, 2771-2780.

Garr E., Bushra B., Tu N. and Delamater A. R. (2019) Goal-directed control on interval schedules does not depend on the action–outcome correlation. Journal of Experimental Psychology: Animal Learning and Cognition, No Pagination Specified.

Geddes C. E., Li H. and Jin X. (2018) Optogenetic Editing Reveals the Hierarchical Organization of Learned Action Sequences. Cell. 174, 3-43.e15.

Gerfen C. R., Paletzki R. and Heintz N. (2013) GENSAT BAC cre-recombinase driver lines to study the functional organization of cerebral cortical and basal ganglia circuits. Neuron. 80, 1368-1383.

Graybiel A. M. (2008) Habits, rituals, and the evaluative brain. Annu. Rev. Neurosci. 31, 359-387.

Gremel C. M., Chancey J. H., Atwood B. K., Luo G., Neve R., Ramakrishnan C., Deisseroth K., Lovinger D. M. and Costa R. M. (2016) Endocannabinoid Modulation of Orbitostriatal Circuits Gates Habit Formation. Neuron. 90, 1312-1324.

Gremel C. M. and Costa R. M. (2013) Orbitofrontal and striatal circuits dynamically encode the shift between goal-directed and habitual actions. Nat. Commun. 4, 2264.

He Y., Li Y., Chen M., Pu Z., Zhang F., Chen L., Ruan Y., Pan X., He C., Chen X., Li Z. and Chen J. F. (2016) Habit Formation after Random Interval Training Is Associated with Increased Adenosine A2A Receptor and Dopamine D2 Receptor Heterodimers in the Striatum. Front. Mol. Neurosci. 9, 151.

Hilario M. R., Clouse E., Yin H. H. and Costa R. M. (2007) Endocannabinoid signaling is critical for habit formation. Front. Integr. Neurosci. 1, 6.

Jenrette T. A., Logue J. B. and Horner K. A. (2019) Lesions of the Patch Compartment of Dorsolateral Striatum Disrupt Stimulus-Response Learning. Neuroscience. 415, 161-172.

Jin X. and Costa R. M. (2015) Shaping action sequences in basal ganglia circuits. Curr. Opin. Neurobiol. 33, 188-196.

Jin X., Tecuapetla F. and Costa R. M. (2014) Basal ganglia subcircuits distinctively encode the parsing and concatenation of action sequences. Nat. Neurosci. 17, 423-430.

Johnston J. G., Gerfen C. R., Haber S. N. and van der Kooy D. (1990) Mechanisms of striatal pattern formation: conservation of mammalian compartmentalization. Brain Res. Dev. Brain Res. 57, 93-102.

Knutson B., Delgado M. R. and Phillips P. E. M. (2009) Chapter 25 - Representation of Subjective Value in the Striatum. Neuroeconomics, 389-406.

Li Y., Pan X., He Y., Ruan Y., Huang L., Zhou Y., Hou Z., He C., Wang Z., Zhang X. and Chen J. F. (2018) Pharmacological Blockade of Adenosine A2A but Not A1 Receptors Enhances Goal-Directed Valuation in Satiety-Based Instrumental Behavior. Front. Pharmacol. 9, 393.

Lingawi N. W., Dezfouli A. and Balleine B. W. (2016) The Psychological and Physiological Mechanisms of Habit Formation. The Wiley Handbook on the Cognitive Neuroscience of Learning, 409-441.

Malvaez M., Greenfield V. Y., Matheos D. P., Angelillis N. A., Murphy M. D., Kennedy P. J., Wood M. A. and Wassum K. M. (2018) Habits Are Negatively Regulated by Histone Deacetylase 3 in the Dorsal Striatum. Biol. Psychiatry. 84, 383-392.

Matsumoto N., Hanakawa T., Maki S., Graybiel A. M. and Kimura M. (1999) Role of [corrected] nigrostriatal dopamine system in learning to perform sequential motor tasks in a predictive manner. J. Neurophysiol. 82, 978-998.

Morigaki R. and Goto S. (2016) Putaminal Mosaic Visualized by Tyrosine Hydroxylase Immunohistochemistry in the Human Neostriatum. Front. Neuroanat. 10, 34.

Nelson A. J. and Killcross S. (2013) Accelerated habit formation following amphetamine exposure is reversed by D1, but enhanced by D2, receptor antagonists. Front. Neurosci. 7, 76.

Nelson A. and Killcross S. (2006) Amphetamine exposure enhances habit formation. J. Neurosci. 26, 3805-3812.

O'Hare J. K., Ade K. K., Sukharnikova T., Van Hooser S. D., Palmeri M. L., Yin H. H. and Calakos N. (2016) Pathway-Specific Striatal Substrates for Habitual Behavior. Neuron. 89, 472-479.

Okada K., Nishizawa K., Fukabori R., Kai N., Shiota A., Ueda M., Tsutsui Y., Sakata S., Matsushita N. and Kobayashi K. (2014) Enhanced flexibility of place discrimination learning by targeting striatal cholinergic interneurons. Nat. Commun. 5, 3778.

Renteria R., Baltz E. T. and Gremel C. M. (2018) Chronic alcohol exposure disrupts top-down control over basal ganglia action selection to produce habits. Nat. Commun. 9, 21-9.

Sieburg M. C., Ziminski J. J., Margetts-Smith G., Reeve H., Brebner L. S., Crombag H. S. and Koya E. (2019) Reward devaluation attenuates cue-evoked sucrose seeking and is associated with the elimination of excitability differences between ensemble and non-ensemble neurons in the nucleus accumbens. eNeuro.

Smith J. B., Klug J. R., Ross D. L., Howard C. D., Hollon N. G., Ko V. I., Hoffman H., Callaway E. M., Gerfen C. R. and Jin X. (2016) Genetic-Based Dissection Unveils the Inputs and Outputs of Striatal Patch and Matrix Compartments. Neuron. 91, 1069-1084.

Stickgold R. and Walker M. P. (2007) Sleep-dependent memory consolidation and reconsolidation. Sleep Med. 8, 331-343.

Van den Bercken, J H and Cools A. R. (1982) Evidence for a role of the caudate nucleus in the sequential organization of behavior. Behav. Brain Res. 4, 319-327.

Yin H. H. (2010) The sensorimotor striatum is necessary for serial order learning. J. Neurosci. 30, 14719-14723.

Yin H. H. and Knowlton B. J. (2006) The role of the basal ganglia in habit formation. Nat. Rev. Neurosci. 7, 464-476.

Yin H. H., Knowlton B. J. and Balleine B. W. (2005) Blockade of NMDA receptors in the dorsomedial striatum prevents action-outcome learning in instrumental conditioning. Eur. J. Neurosci. 22, 505-512.

Yin H. H., Knowlton B. J. and Balleine B. W. (2004) Lesions of dorsolateral striatum preserve outcome expectancy but disrupt habit formation in instrumental learning. Eur. J. Neurosci. 19, 181-189.

Yu C., Gupta J., Chen J. F. and Yin H. H. (2009) Genetic deletion of A2A adenosine receptors in the striatum selectively impairs habit formation. J. Neurosci. 29, 15100-15103.

---

## [Decision Letter · Decision Letter 1]

18 Dec 2019

PONE-D-19-28957R1

Lesion of striatal patches disrupts habitual behaviors and increases behavioral variability

PLOS ONE

Dear Mr. Nadel,

Thank you for submitting your manuscript to PLOS ONE. After careful consideration, we feel that it has merit but does not fully meet PLOS ONE’s publication criteria as it currently stands. Therefore, we invite you to submit a revised version of the manuscript that addresses the points raised during the review process.

The reviewers are mostly satisfied with the submitted revision, though two minor lingering points need to be addressed. (1) the genetic mutation is not entirely specific to striosomes and is 'leaky.' While this in no way diminishes the value of the reported findings, this needs to clearly be stated in the discussion as a caveat with a very brief description how not completely specific to striosomes, (2) another reviewer continues to request raw data. As this does not seem to be a concern for the other reviewers, I would ask the authors provide me (editor) an explanation of why raw data is not included. The manuscript will not need to go out for review again. The authors' responses will be evaluated promptly by the editor and a final decision rendered.

We would appreciate receiving your revised manuscript by Feb 01 2020 11:59PM. To enhance the reproducibility of your results, we recommend that if applicable you deposit your laboratory protocols in protocols.io, where a protocol can be assigned its own identifier (DOI) such that it can be cited independently in the future. For instructions see: http://journals.plos.org/plosone/s/submission-guidelines#loc-laboratory-protocols

We look forward to receiving your revised manuscript.

Kind regards,

Jeff A Beeler

Academic Editor

PLOS ONE

Reviewers' comments:

Reviewer's Responses to Questions

**Comments to the Author**

1. If the authors have adequately addressed your comments raised in a previous round of review and you feel that this manuscript is now acceptable for publication, you may indicate that here to bypass the “Comments to the Author” section, enter your conflict of interest statement in the “Confidential to Editor” section, and submit your "Accept" recommendation.

Reviewer #1: All comments have been addressed

Reviewer #2: (No Response)

Reviewer #3: All comments have been addressed

2. Is the manuscript technically sound, and do the data support the conclusions?

Reviewer #1: Yes

Reviewer #2: Partly

Reviewer #3: Partly

3. Has the statistical analysis been performed appropriately and rigorously? 

Reviewer #1: Yes

Reviewer #2: Yes

Reviewer #3: Yes

4. Have the authors made all data underlying the findings in their manuscript fully available?

Reviewer #1: Yes

Reviewer #2: Yes

Reviewer #3: Yes

5. Is the manuscript presented in an intelligible fashion and written in standard English?

Reviewer #1: Yes

Reviewer #2: Yes

Reviewer #3: Yes

6. Review Comments to the Author

Reviewer #1: My concerns have been addressed by the authors. the manuscript has been improved considerably. The discussion now includes better acknowledgment of previous work, including the difference between omission and devaluation as assays for habitual performance. The additional discussion of different aspects of habitual behavior is also appropriate. I think the paper is now publishable.

Reviewer #2: Overall, the authors have done a good job at addressing my previous comments. My additional comments are listed below.

- Regarding my primary concern about the devaluation procedure, the authors claim that “the current approach using home-cage chow and liquid sucrose has been successfully employed in the literature (He et al. 2016; Li et al. 2018)”. The results of the He et al. paper alleviate my concern because the mice are food-deprived and when they are given selective satiety tests after training on a random ratio schedule, they show instrumental sensitivity to devaluation. This is encouraging. The Li et al. paper is less relevant because the mice are not reported as food-deprived, so I recommend removing this citation. The authors also appeal to the fact that other researches have used a combination of liquid and solid reinforcers during devaluation tests, but this does not relate to my concern and distracts from the main issue. My concern is a higher rate of consumption during devalued vs. valued tests (I would be concerned about this even if the authors used wildtype mice, so translating this concern into a caveat should generalize beyond Sepw1 mice). If the assignment of solid and liquid reinforcers is balanced across subjects, this concern goes away (mostly). I strongly recommend the authors rephrase the following paragraph by removing the first sentence and amending the second sentence to mean mice in general (not just Sepw1 mice):

“The combination of liquid and solid reinforcers used across devaluation and valuation trials is common (Gremel and Costa 2013; Gremel et al. 2016; He et al. 2016; Li et al. 2018; Renteria et al. 2018). Nevertheless, the lack difference in our devaluation and valuation trials could be attributed to this asymmetry in reward, which Sepw1 mice may be particularly sensitive to”.

- I previously recommended that the authors display and analyze the raw pressing rates from the selective satiety tests. However, in the revised paper the authors have only presented and analyzed raw pressing rates from the devalued condition but not the valued condition. I apologize if I was not clear, but this is the standard way of analyzing devaluation test data (comparing valued vs. devalued for each group). Without the raw pressing rates from the valued condition, there is no way to compare figures 4A and 4C. If the authors are going to present the raw pressing rates from the devalued condition, they must also present the raw pressing rates from the valued condition.

Reviewer #3: The authors have largley addressed my concerns. The main one being the leakiness of the line. It still feels a bit misleading to state investigation of a patch deletion on these behaviors. The question of patch like-due to development is interesting, but a separate investigation.

7. PLOS authors have the option to publish the peer review history of their article (what does this mean?). If published, this will include your full peer review and any attached files.

Reviewer #1: No

Reviewer #2: No

Reviewer #3: No

---

## [Author Response · Author response to Decision Letter 1]

18 Dec 2019

Dear Dr. Beeler,

We would like to again thank you and the reviewers for insightful comments and prompt processing of our manuscript. Below we address each comment raised by the reviewers. We have created two new panels in Figure 4, and have updated our Graphpad Prism file accordingly. Moreover, we have included new discussion and edited wording in several places in the document. Again, we feel the work has been made stronger through this process. Below you will find our responses to each concern with reviewer comments italicized, changes to text in bold, and our own comments in plain text. 

Reviewer #1: My concerns have been addressed by the authors. the manuscript has been improved considerably. The discussion now includes better acknowledgment of previous work, including the difference between omission and devaluation as assays for habitual performance. The additional discussion of different aspects of habitual behavior is also appropriate. I think the paper is now publishable.

We again thank the reviewer for their constructive feedback on this work, we agree that the manuscript has benefited significantly from the inclusion of these topics. 

Reviewer #2: Overall, the authors have done a good job at addressing my previous comments. My additional comments are listed below.

- Regarding my primary concern about the devaluation procedure, the authors claim that “the current approach using home-cage chow and liquid sucrose has been successfully employed in the literature (He et al. 2016; Li et al. 2018)”. The results of the He et al. paper alleviate my concern because the mice are food-deprived and when they are given selective satiety tests after training on a random ratio schedule, they show instrumental sensitivity to devaluation. This is encouraging. The Li et al. paper is less relevant because the mice are not reported as food-deprived, so I recommend removing this citation. 

This citation has been removed.

The authors also appeal to the fact that other researches have used a combination of liquid and solid reinforcers during devaluation tests, but this does not relate to my concern and distracts from the main issue. My concern is a higher rate of consumption during devalued vs. valued tests (I would be concerned about this even if the authors used wildtype mice, so translating this concern into a caveat should generalize beyond Sepw1 mice). If the assignment of solid and liquid reinforcers is balanced across subjects, this concern goes away (mostly). I strongly recommend the authors rephrase the following paragraph by removing the first sentence and amending the second sentence to mean mice in general (not just Sepw1 mice):

“The combination of liquid and solid reinforcers used across devaluation and valuation trials is common (Gremel and Costa 2013; Gremel et al. 2016; He et al. 2016; Li et al. 2018; Renteria et al. 2018). Nevertheless, the lack difference in our devaluation and valuation trials could be attributed to this asymmetry in reward, which Sepw1 mice may be particularly sensitive to”.

We apologize for the confusion on this point. This has been done. Lines 458-465 now read:

“First, the use of home-cage chow and liquid sucrose rewards could represent an asymmetrical manipulation between devaluation and valuation probe trials, which may have impacted the results of these probe trials. However, the approach used here has been utilized in a previous study, and these mice remained sensitive to devaluation [44]. Nevertheless, the lack of difference in our probes could be attributed to potential asymmetry in consumption before devaluation and valuation probes.”

- I previously recommended that the authors display and analyze the raw pressing rates from the selective satiety tests. However, in the revised paper the authors have only presented and analyzed raw pressing rates from the devalued condition but not the valued condition. I apologize if I was not clear, but this is the standard way of analyzing devaluation test data (comparing valued vs. devalued for each group). Without the raw pressing rates from the valued condition, there is no way to compare figures 4A and 4C. If the authors are going to present the raw pressing rates from the devalued condition, they must also present the raw pressing rates from the valued condition.

We apologize for the confusion here as well. We have now performed this analysis and included two new panels in Figure 4. As shown below, presses between devaluation and valuation trials were not different in either control or lesioned groups (A-B). This result has been updated in Results (Line 305-307), Figure legend 4 (Line 321-322), and the paired student’s t-test used to compare these is now in Statistical Analysis (Line 194-195).

Reviewer #3: The authors have largley addressed my concerns. The main one being the leakiness of the line. It still feels a bit misleading to state investigation of a patch deletion on these behaviors. The question of patch like-due to development is interesting, but a separate investigation.

The author highlights an important limitation of the current work. While the Sepw1 line has been used in many other studies to determine patch function (Crittenden et al. 2016; Smith et al. 2016; Yoshizawa et al. 2018), the off-site targeting of exopatch neurons remains problematic. Other Cre lines are available and provide viable alternatives, including Mash1-CreER (Bloem et al. 2017), 599-CreER (McGregor et al. 2019), or Oprm1-Cre (Märtin et al. 2019), though each of these lines suffers from some level of exopatch Cre expression (or expression of Cre in matrix). In acknowledgment of this shortcoming, we have added the following paragraph to discussion:

“While Sepw1 NP67 mice have preferential expression of Cre recombinase in patch projection neurons, a limitation of the current work is the expression of Cre in ‘exo-patch’ neurons (Smith et al 2017), resulting in the lesioning of both patch and exo-patch neurons (Figure 1). Exo-patch neurons have similar gene expression and connectivity profiles to patch neurons (Smith et al 2017), but they fall outside of traditionally defined patches (Crittenden et al 2011). The Sepw1 NP67 line has been previously used to study patch connectivity (Smith et al, Crittenden et al 2016) and activity (Yoshizawa et al 2018). Other recent studies have utilized alternative Cre lines to target patches, including Mash1-CreER (Bloem et al 2017), 599-CreER (McGregor et al 2019), or Oprm1-Cre (Märtin et al 2019), though each of these lines also has some off-target labeling of exo-patch or matrix neurons. Thus, while the current work suggests lesions preferentially targeting patches impair aspects of habitual behavior, it remains unknown what role exo-patch neurons play in this process.”

References

Bloem B., Huda R., Sur M. and Graybiel A. M. (2017) Two-photon imaging in mice shows striosomes and matrix have overlapping but differential reinforcement-related responses. Elife. 6, 10.7554/eLife.32353.

Crittenden J. R., Tillberg P. W., Riad M. H., Shima Y., Gerfen C. R., Curry J., Housman D. E., Nelson S. B., Boyden E. S. and Graybiel A. M. (2016) Striosome-dendron bouquets highlight a unique striatonigral circuit targeting dopamine-containing neurons. Proc. Natl. Acad. Sci. U. S. A. 113, 11318-11323.

Märtin A., Calvigioni D., Tzortzi O., Fuzik J., Wärnberg E. and Meletis K. (2019) A Spatiomolecular Map of the Striatum. bioRxiv, 613596. (Preprint)

McGregor M. M., McKinsey G. L., Girasole A. E., Bair-Marshall C. J., Rubenstein J. L. R. and Nelson A. B. (2019) Functionally Distinct Connectivity of Developmentally Targeted Striosome Neurons. Cell. Rep. 29, 1419-1428.e5.

Smith J. B., Klug J. R., Ross D. L., Howard C. D., Hollon N. G., Ko V. I., Hoffman H., Callaway E. M., Gerfen C. R. and Jin X. (2016) Genetic-Based Dissection Unveils the Inputs and Outputs of Striatal Patch and Matrix Compartments. Neuron. 91, 1069-1084.

Yoshizawa T., Ito M. and Doya K. (2018) Reward-Predictive Neural Activities in Striatal Striosome Compartments. eNeuro. 5, 10.1523/ENEURO.0367-Feb.

Again, we thank you and the reviewers for careful consideration of our work. We look forward to re-review of our manuscript.

We would be grateful to be considered in the Neuroscience of Reward and Decision Making Call for Papers and we have no changes to our financial disclosure.

---

## [Editor Report · Decision Letter 2]

23 Dec 2019

Lesion of striatal patches disrupts habitual behaviors and increases behavioral variability

PONE-D-19-28957R2

Dear Dr. Nadel,

We are pleased to inform you that your manuscript has been judged scientifically suitable for publication and will be formally accepted for publication once it complies with all outstanding technical requirements.

With kind regards,

Jeff A Beeler

Academic Editor

PLOS ONE

Additional Editor Comments (optional):

Congratulations. You have succeeded in a rigorous review process. I look forward to publication of your paper!
---

## [Editor Report · Acceptance letter]

30 Dec 2019

PONE-D-19-28957R2 

Lesion of striatal patches disrupts habitual behaviors and increases behavioral variability 

Dear Dr. Nadel:

I am pleased to inform you that your manuscript has been deemed suitable for publication in PLOS ONE. Congratulations! Your manuscript is now with our production department. 

With kind regards,

on behalf of

Dr. Jeff A Beeler 

Academic Editor

PLOS ONE